# Contributions of *Zea mays* subspecies *mexicana* haplotypes to modern maize

Ning Yang[1], Xi-Wen Xu[1], Rui-Ru Wang[1], Wen-Lei Peng[1], Lichun Cai[2], Jia-Ming Song[1], Wenqiang Li[1], Xin Luo[1], Luyao Niu[1], Yuebin Wang[1], Min Jin[1], Lu Chen[1], Jingyun Luo[1], Min Deng[1], Long Wang[1], Qingchun Pan[1], Feng Liu[1], David Jackson [1,3], Xiaohong Yang[2], Ling-Ling Chen[1] & Jianbing Yan[1]

Maize was domesticated from lowland teosinte (*Zea mays* ssp. *parviglumis*), but the contribution of highland teosinte (*Zea mays* ssp. *mexicana*, hereafter *mexicana*) to modern maize is not clear. Here, two genomes for Mo17 (a modern maize inbred) and *mexicana* are assembled using a meta-assembly strategy after sequencing of 10 lines derived from a maize-teosinte cross. Comparative analyses reveal a high level of diversity between Mo17, B73, and *mexicana*, including three Mb-size structural rearrangements. The maize spontaneous mutation rate is estimated to be $2.17 \times 10^{-8}$ ~$3.87 \times 10^{-8}$ per site per generation with a nonrandom distribution across the genome. A higher deleterious mutation rate is observed in the pericentromeric regions, and might be caused by differences in recombination frequency. Over 10% of the maize genome shows evidence of introgression from the *mexicana* genome, suggesting that *mexicana* contributed to maize adaptation and improvement. Our data offer a rich resource for constructing the pan-genome of *Zea mays* and genetic improvement of modern maize varieties.

[1] National Key Laboratory of Crop Genetic Improvement, Huazhong Agricultural University, Wuhan 430070, China. [2] National Maize Improvement Center of China, Beijing Key Laboratory of Crop Genetic Improvement, China Agricultural University, Beijing 100193, China. [3] Cold Spring Harbor Laboratory, Cold Spring Harbor, New York 11724, USA. Ning Yang and Xi-Wen Xu contributed equally to this work. Correspondence and requests for materials should be addressed to L.-L.C. (email: llchen@mail.hzau.edu.cn) or to J.Y. (email: yjianbing@mail.hzau.edu.cn)

Maize (*Zea mays* ssp. *mays*) is one of the most important crops globally, with an annual production of >1 billion tons[1], and has been a model for biological studies for over a century. It is also an incredibly diverse species as up to half of the genome is not shared between any two maize varieties[2, 3]. Structural variations (SVs) including copy number variations (CNVs), presence/absence variations (PAVs), inversions, and translocations contribute to the diversity of the maize genome[4], and also play an important role in phenotypic variation[5, 6]. Multiple reference genomes are, therefore, required in order to represent all of the genomic content for maize and to understand its domestication, adaptation, and improvement. On the basis of available archaeological and molecular evidence, it is believed that maize domestication started approximately 9000 years ago in Southwest Mexico, from the lowland teosinte *Zea mays* ssp. *parviglumis* (hereafter, *parviglumis*)[7, 8]. *Zea mays* ssp. *mexicana* (hereafter, *mexicana*) is distributed across the cooler higher elevations of the Mexican Central Plateau, and can easily cross with maize[9], such that gene flow from *mexicana* has contributed to maize local adaptation and improvement[8, 10]. However, this process has not been investigated on a genome-wide basis, because no *mexicana* genome was available up to now[10, 11].

In this study, a meta-assembly strategy combining a new genetic design (Fig. 1a, b), based on selections from an inter-specific cross between Mo17 and *mexicana* was used to assemble the contigs and scaffolds of the parental maize *mexicana* genomes, and the scaffolds were anchored with genetic map and publicly available genotyping-by-sequencing data[6]. This design allowed us to estimate the maize and teosinte spontaneous point mutation rate in each generation based on the identity by descent (IBD) regions of ten sequenced individuals. We compared large structural variations among *Zea* genomes, and hundreds of genes under positive selection were identified. In addition, the introgression of *mexicana* to maize was investigated. All the above analysis provides a resource for understanding maize adaptation history and the value of its wild relatives.

## Results

**Genome assembly via a new genetic design.** A selfed backcross (BC$_2$F$_7$) population (hereafter, TM population) was derived from a single F$_1$ cross between Mo17 and *mexicana*, and consisted of 191 individuals, which were genotyped with the Illumina 56 K MaizeSNP50 array[12] (Fig. 1a). Ten selected individuals covering

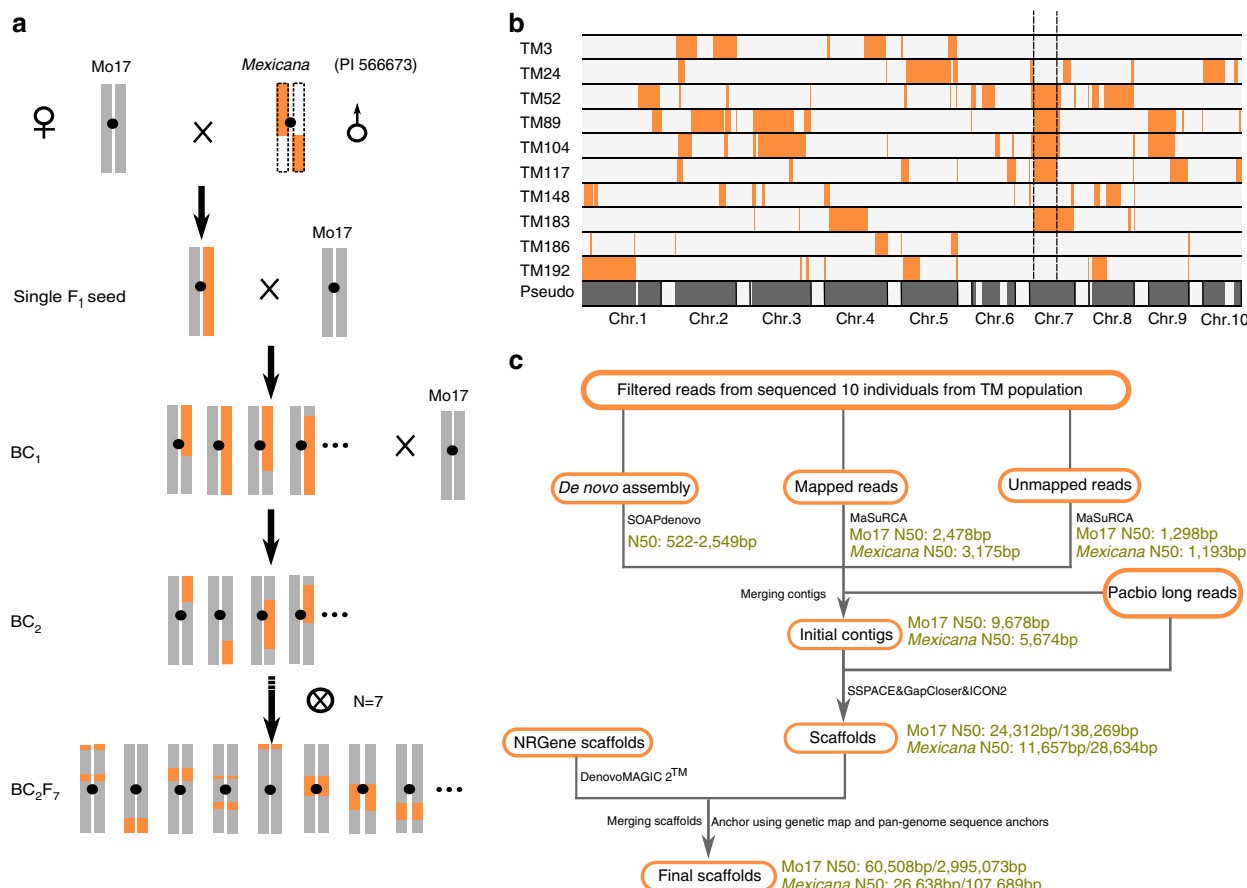

**Fig. 1** Mo17-*mexicana* population development and characterization of the assembled genomes. **a** The construction of Mo17-*mexicana* BC$_2$F$_7$ population. Only a single F$_1$ seed was used. **b** The genetic design for meta-assembly of the Mo17 and *mexicana* genomes. Ten well-chosen TM lines that genetically divided the Mo17 and *mexicana* genomes into different fragments. The fragment indicated by the dash lines are used to estimate the mutation rate in *mexicana*. **c** Schematic illustration of the meta-assembly pipeline. The Illumina reads were filtered and adapters were removed for each library and the assembly can be divided into three major steps: Step 1, three strategies were used to assemble contigs, including de novo assembly of 10 individuals, reference-based assembly based on B73 genome and de novo assembly of unmapped reads. Step 2, the contigs obtained from the three strategies were merged, and further connected or extended with Pacbio long reads. Long mate-pair libraries were then added to assemble scaffolds. Step 3, the NRGene scaffolds were integrated, and the merged scaffolds were the final scaffolds. The number represents contig length before diagonal and scaffold length after diagonal

**Table 1 Characteristics of the Mo17 and *mexicana* genome sequences**

|  | Mo17 | *mexicana* |
|---|---|---|
| Length of assembled scaffolds (bp) | 2,041,547,554 | 1,204,281,382 |
| Number of scaffolds ≥ 500 bp | 48,268 | 107,418 |
| Largest scaffolds (bp) | 26,086,894 | 26,384,332 |
| Scaffold N50 (bp) | 2,995,073 | 107,689 |
| Length of assembled contigs (bp) | 2,005,908,899 | 1,157,520,532 |
| Number of contigs ≥ 500 bp | 116,972 | 149,360 |
| Largest contig (bp) | 564,568 | 564,568 |
| Contig N50 (bp) | 60,508 | 26,638 |
| Sequences anchored to chromosomes (%) | 98.4% | 95.9% |
| Numbers of gene models/transcripts | 40,003/97,069 | 31,387/71,535 |
| Mean transcript length (bp) | 2219.67 | 2065.46 |
| Mean coding sequence length (bp) | 1310.16 | 1255.75 |
| Total size of transposable elements (bp) | 1,580,816,259 | 832,324,464 |

the whole Mo17 genome and ~96% of the *mexicana* genome were sequenced, and the Mo17 and *mexicana* genomes were represented by 211 and 176 distinct chromosomal segments, respectively (Fig. 1b and Supplementary Data 1, Supplementary Data 2). A meta-assembly strategy (Fig. 1c) integrating contigs from Illumina reads (1.17 TB, Supplementary Table 1), PacBio long reads (9.29 GB, Supplementary Table 2), and DenovoMAGIC 2[TM] (445.83 GB, Supplementary Table 3, NRGene, Israel) was adopted to separately assemble Mo17 and *mexicana* genomes (Supplementary Note 1 and Supplementary Fig. 1). The two final assembled genomes were 2.04 Gb for Mo17 and 1.20 Gb for *mexicana*, with scaffold N50s of 3 Mb and 107 kb, respectively (Table 1). We anchored and ordered 1973 Mb (96.6%) and 1072 Mb (88.8%) of the Mo17 and *mexicana* scaffolds with the genetic map of the TM population, and an additional 36.4 Mb (1.8%) of the Mo17 genome and 85.5 Mb (7.1%) of the *mexicana* genome were anchored with the publicly available genotyping-by-sequencing anchors from 14,129 maize inbred lines[6] (Supplementary Note 2).

Curated repeat libraries and de novo prediction were used to annotate the transposable elements (TEs) that comprised 79.7% of Mo17 and 72.8% of the *mexicana* genome (Table 1 and Supplementary Table 4). Long terminal repeat (LTR) elements were the major class of TEs in Mo17 and *mexicana*, similar to the B73 reference genome (Supplementary Fig. 2a, b). A total of 40,003 and 31,387 high-confidence protein-coding gene models were predicted for Mo17 and *mexicana*, by combining *ab initio* prediction and evidence-based approaches (protein, EST, and RNA-seq of three tissues) (Supplementary Table 5 and Supplementary Note 3). Comparative analysis with B73, rice, and sorghum revealed that a core set of 10,878 gene families were shared among all five genomes (Supplementary Fig. 3a). Almost 2,000 conserved protein families were identified among B73, Mo17, and *mexicana* genomes, similar to the comparison of B73 and other grass species (i.e., rice and sorghum)[4, 13, 14].

The quality of the assembled genomes was confirmed using the following approaches: (1) 93% of the Plantae BUSCO[15] data set could be aligned to the assembled Mo17 genome, which was similar to the B73 reference genome[4]. For *mexicana*, 86% of the BUSCO gene set was successfully aligned; (2) 96.0% (88.7%) and 90.7% (83.1%) of the partial (complete) core eukaryotic genes mapping approach (CEGMA) gene set from six model organisms[16] could be mapped to the assembled Mo17 and *mexicana* genomes, with coverage > 70%; (3) the highly conserved core gene families (coreGFs) weighted score[17] for Mo17 and *mexicana* was 94.0% and 87.6%, respectively (Supplementary Table 6); (4)

98.2% and 81.5% of the genes from B73 reference genome[4] (RefGen_v2 release 5b) could be mapped to Mo17 and *mexicana* genomes, respectively, with coverage > 80%; (5) two Mo17 bacterial artificial chromosome (BAC) clones that had previously been sequenced for *Vgt1* (Accession: EF659467) and *ZmWAK* cloning[5, 18] showed alignments of 99.9% and 99.8% with our Mo17 assembled genome (Supplementary Fig. 3b, c), and one Mo17 scaffold sequence[19] also aligned to our Mo17 genome with an identity of 99.9% (Supplementary Fig. 3d); and (6) four Mo17 BACs were randomly selected from a newly constructed BAC library and sequenced on PacBio RSII platform. All the four BACs showed high consistence with our assembly (Supplementary Fig. 3e–h). In summary, these results support the conclusion that the assembled quality of Mo17 is acceptable. Because the sequencing coverage of *mexicana* is much lower than Mo17, the assembled genome length of *mexicana* is not as complete as Mo17, only half of *mexicana* genome was assembled, and most of the un-assembled regions were repetitive sequences. However, the protein-coding regions in Mo17 and *mexicana* genomes are both well assembled (Supplementary Table 6).

**Zea genomes contain large structural variations.** Structural variations are widespread in maize genomes and are important for local adaptation and genetic improvement[20, 21]. Considering that the *mexicana* genome is incomplete, we only compared the presence/absence variations (PAVs, ≥ 100 bp) between B73 and Mo17 by using comparatively genomic and read-mapping methods (Methods section). Consequently, 220,860 (~88.8 Mb) PAVs between B73 and Mo17 were identified (Supplementary Fig. 4a, b and Supplementary Table 7). These PAVs were unevenly distributed across the genome (Fig. 2). The rate of TE-related PAVs (PAVs containing ≥ 80% TE) is ~ 34.6%, more than in other species[22, 23], indicating that they are closely related to transposition events. Full-length LTR retrotransposons in PAV regions were also identified (Supplementary Table 7), indicating recent transposition events. A total of 1,293 PAV genes between B73 and Mo17, were identified, and these PAV genes tended to be shorter than other annotated genes and the underlying mechanism is still unclear (Supplementary Fig. 4c).

Three PAVs larger than 1 Mb were identified between Mo17 and B73. PAVs in gene regions are increasingly considered to be one of the origins of phenotypic variations across/within species[4, 24–27]. For example, PAVs in maize are causal variants for disease resistance[21, 28], including a large one on chromosome 6 between Mo17 and B73, which is putatively responsible for variation in sugarcane mosaic virus resistance[28] (Supplementary Fig. 5a). In the present study, we validated this PAV and found that the deletion in Mo17 was ~ 2.9 Mb, ~ 300 kb larger than previous estimates[21] (Supplementary Table 8 and Supplementary Fig. 5b). In addition, two new large insertions ( > 1 Mb) were identified in Mo17 genome compared with B73. One ~2.2 Mb insertion, containing 10 annotated genes, was located at 69.7–71.9 Mb of Mo17 chromosome 6 (Supplementary Table 9). Consequently, this region could be PCR amplified from Mo17, but not from B73 (Supplementary Table 8 and Supplementary Fig. 5a–c). Another ~1.4 Mb insertion was located at 143.6–145.1 Mb of Mo17 chromosome 4, and contained 8 annotated genes (Supplementary Table 9) that were also validated by PCR amplification (Supplementary Table 8 and Supplementary Fig. 5d, e). Six of the 18 annotated genes within these two large insertions were expressed in one or more tissues (Supplementary Table 9); however, the phenotypic effects of these two large PAVs remain to be established.

An inversion (*Inv9d*) showing no crossovers within a ~30 Mb region on the short arm of chromosome 9 was identified between

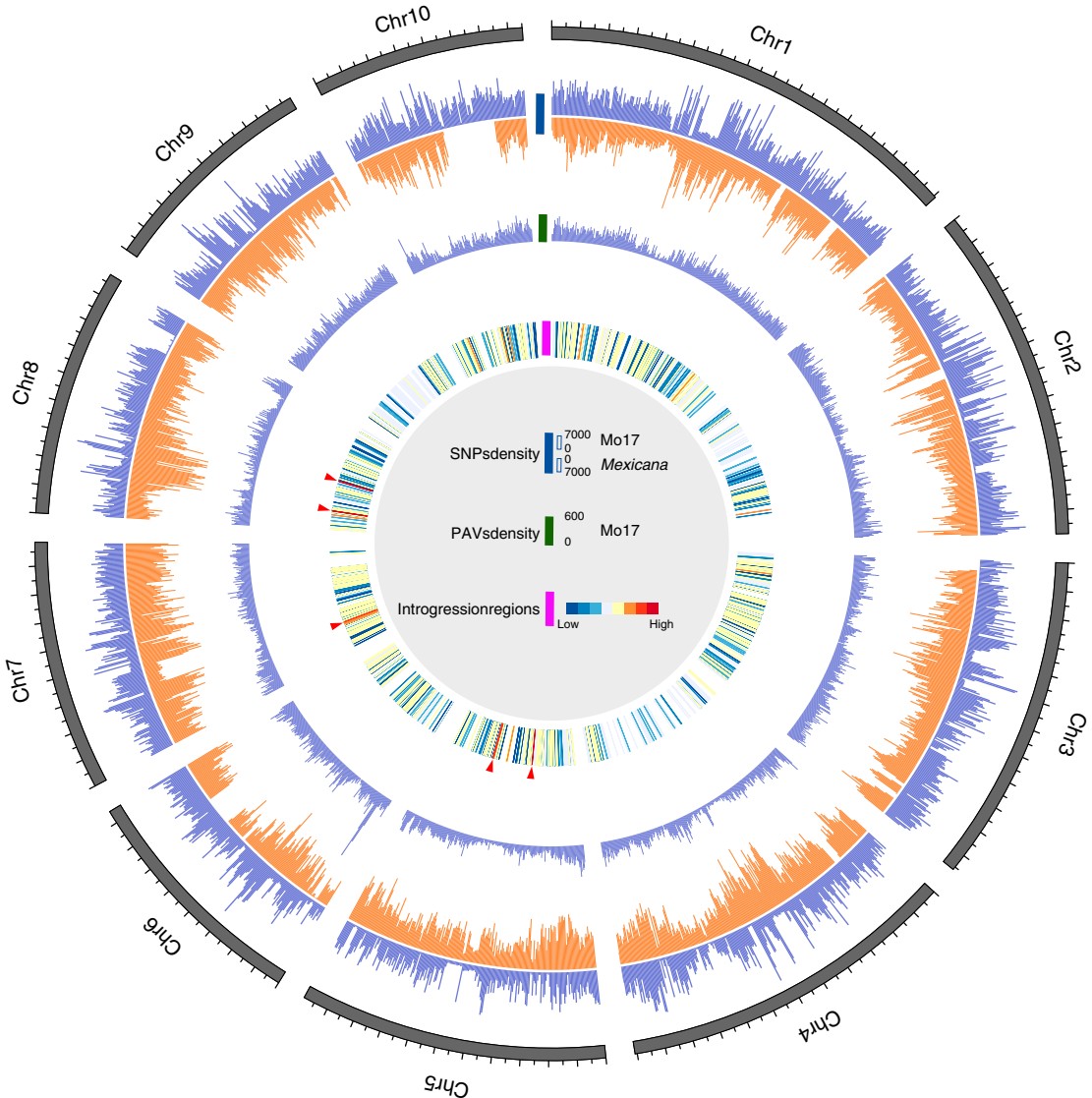

**Fig. 2** Comparative analysis of Mo17 and *mexicana* genomes. Tracks from outer to inner circles indicate: SNPs, PAV density (window size of 1 Mb, the outer and inner layers indicate Mo17 (Blue) and *mexicana* (Orange)respectively), and a heatmap of introgression, indicating number of lines containing the corresponding introgression region (low = blue; high = red), hotspots are labeled with red arrows

modern maize (B73/Mo17) and *mexicana*, and was confirmed by the TM genetic map and linkage disequilibrium (LD) analysis based on 95 *mexicana* accessions genotyped with the Illumina MaizeSNP50 array[12] (Fig. 3a, b). This inversion was found only in some *mexicana* accessions and not in *parviglumis* in a previous study[20]. The *Inv9d* homologous region of B73 (~27 Mb) was shorter than that of *mexicana* (~38 Mb); therefore, a distinct arc in the synteny dot-plot was observed within the inversion region, and the gradient of gene density increase towards chromosome ends was disrupted in the B73 homologous region (Fig. 3c). This phenomenon may be caused by changes in rates of DNA loss and gain in regions that switch from chromosome ends to the near pericentromeric contexts[29]. We further compared the gene co-linearity between rice and *mexicana*/B73 using MCScan[30, 31] and found that the *Inv9d* region showed a well co-linearity between *mexicana* and rice in the synteny block, whereas there was an inversion between B73 and rice (Fig. 3d, e). Rice has largely preserved the ancestral karyotype of the grass common ancestor, with no major changes in genome structure after it separated from other clades, and has been least affected by transposon activities and accumulation[30]. It implied that the *mexicana*

genome was closer to the ancestor state and the inversion in modern maize genome was derived from selection. To explore the biological impact of *Inv9d*, we found a large ear leaf width QTL within the *Inv9d* region (Fig. 3b, f), which was consistently identified in six environments in our TM population. The inversion causes a wider ear leaf width that can increase the leaf area, thus increasing the photosynthesis. Furthermore, *Inv9d* showed altitudinal clines in environmental association analysis in a previous study[20], emphasizing its potential role in local adaptation to ecological factors. However, further efforts are needed to identify the underlying genes and possible application value in maize breeding.

**Mexicana genome contributes to maize adaptation**. It is interesting to ask how maize domesticated from lowland *parviglumis* and adapted to highland environments. This may have been facilitated by gene flow (introgression) from highland *mexicana*[8]. Combining our new genomic resource and the hapmap3[32] of maize, the introgression region in each individual varied from 0.005% to 0.724% as determined by investigation of 895 inbreds,

and this is likely to be an underestimate (Supplementary Fig. 6b), since marker density was low (Fig. 2) (Methods section). By summing the unique introgression regions from all investigated individuals, we found that as much as 10.7% of the maize genome showed evidence of introgression from *mexicana*. However, based on a comparison of the current genome data (B73, Mo17 and *mexicana*), the B73 and Mo17 genome only contained ~1.2% (~24.5 Mb) and ~0.34% (~7.1 Mb) of putative introgression regions from *mexicana*, respectively (Fig. 2) (Methods section). These results could provide a better resolution as compared with

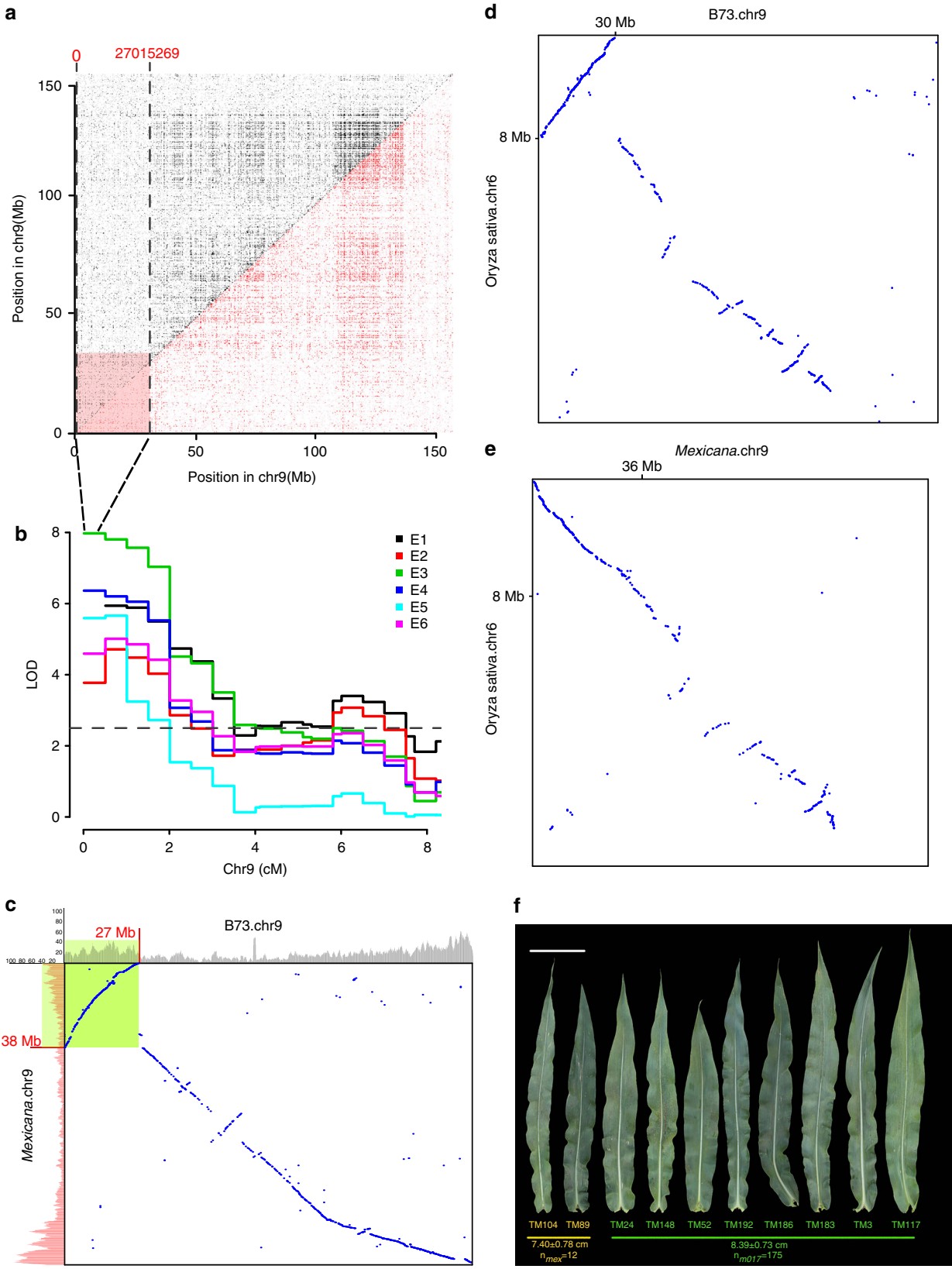

a previous estimation[10] (Supplementary Data 3). Several introgression hot spots were identified, and one of them located on chromosome 8 and is shared among 312 lines (Fig. 2). However, nearly half of the introgressed regions are specific to a given inbred (Supplementary Fig. 6c). Leaf macrohairs and pigment intensity are regarded as adaptive traits under highland conditions, and QTLs for those traits have been identified using a segregating *parviglumis-mexicana* population[33]. At least three of these adaptation traits QTLs overlap with introgression regions were identified in the present study (Supplementary Data 4). The highland Mexican maize have more sympatric populations with *mexicana*, and their expected introgression proportion should be higher than that in lowland maize. A new genotype by sequencing (GBS) data set of 4,022 landraces[34] in which 2900 landraces had their elevation information were further used to assess the introgression proportion in highland Mexican maize. We found that maize from highland indeed have significantly higher introgression proportion than that in lowland Mexico (Supplementary Fig. 7a, $P = 0.002$). Furthermore, we found no matter where the maize came from (Mexico, Honduras, Guatemala, Panama Costa Rica, etc), the introgression proportion was significantly correlated ($P = 9.703e{-}06$) with the elevation ($n = 482$) (Supplementary Fig. 7b). This result indicated that some introgression regions should associate with highland adaptation, and had been kept when maize expand to other places.

**Patterns of spontaneous mutations in the maize genome.** Mutation is the foundation of evolution and the origin of phenotypic variation in populations[35]. The design of this study allowed a direct estimate of the spontaneous mutation rate on a genomic scale, using the Mo17 segments shared among the 10 selected TM individuals with 10 continuous generation selection inherited from a single macrospore (Fig. 1a, b). In total, 7960 spontaneous point mutations were identified and the mutation rate was estimated to be $3.87 \times 10^{-8}$ per site per generation, using strict criteria of excluding all possible regions of residual heterozygosity (Supplementary Fig. 8a, b and Methods section). For further excluding the influences of remaining heterozygosity and/or different Mo17 individual being used for developing TM population, a large *mexicana* fragment on chromosome 7 (38,378,754–105,766,898 bp) covered by 5 individuals were selected to estimate the mutation rate in *mexicana* (Fig. 1b). Using the same filtering criterions, we estimated the mutation rate of *mexicana* was $2.17 \times 10^{-8}$ (73 point mutations in ~67 Mb region). Therefore, we estimated the spontaneous mutation rate in the maize genome is $2.17 \sim 3.87 \times 10^{-8}$, which is slightly higher than the previous indirect estimation using IBD regions in modern maize breeding ($1.63 \times 10^{-8}$)[36], and is also higher than the mutation rate in *Arabidopsis thaliana* ($5.9 \sim 7.1 \times 10^{-9}$)[37].

The global landscape of mutations demonstrated a remarkably nonrandom distribution in the genome ($P = 1.2 \times 10^{-20}$, KS test, Fig. 4a), a phenomenon also observed in viruses, bacteria, and many eukaryotes, including yeast, humans, and *Arabidopsis*[38, 39]. At the chromosome level, the mutation rate had weak correlation with chromosome length ($P = 0.02$, r = 0.86), and chromosome 2 seemed to have a higher mutation rate than others

(Supplementary Table 10). The mutation rate was about 3.5 fold higher in genic regions ($8.84 \times 10^{-8}$, including upstream regions (−5 kb), downstream regions ( + 5 kb), UTRs, exons, and introns) than in the intergenic regions ($2.50 \times 10^{-8}$) (Fig. 4c and Supplementary Table 11). A nonrandom distribution was also observed within the genic regions, with the highest mutation rate in the gene bodies, and the upstream regions had a higher mutation rate than the downstream regions[40] (Fig. 4b and Supplementary Fig. 8c). A lower mutation rate was observed in the hypermethylated centromere regions[41, 42], opposite to the findings in *Arabidopsis*[37]. Hypermethylation may affect the mutation rate, but may be masked by other effects, such as recombination, especially in the chromosome arms where recombination is promoted. In coding regions, amino acid substitutions are expected to have a large impact on the biological function of the protein. In total, 279 new mutations were found in protein-coding regions (nonsynonymous/synonymous = 151/ 128), including 37 predicted deleterious mutations[43]. The mutation rate of $3.18 \times 10^{-8}$ (19 mutations) in the protein-coding regions of pericentromeric genes was lower than in the chromosome arms ($4.55 \times 10^{-8}$, 260 mutations). However, the ratio of nonsynonymous to synonymous changes was higher in the pericentromeric regions ($11/8\approx1.38$) than that in the chromosome arms ($140/120\approx1.17$), and the deleterious mutation rate in the pericentromeric regions was about two-fold higher ($5/19\approx0.26$ vs. $32/260\approx0.12$) (Fig. 4c). Deleterious mutations may be more common in the pericentromeres due to the lower recombination rate, which reduces their possibility of elimination[44, 45]. A context-dependent effect[44, 46], namely the dependence of the substitution rate at a site on two flanking nucleotides, was also observed (Fig. 4d). The triplets "ASA" and "CSG" (S = "C" or "G") were more common (1.7-fold more than the mean of other triplets), and may be explained by context-dependent DNA replication errors, cytosine deamination, or biased gene conversion[46]. The fact that CG is the most common type of methylation may also explain part of this phenomenon[41]. Transitions were 2.34 times more frequent than transversions, and GC > AT transition was the most frequent (Fig. 4e). However, the rate of transitions/transversions in genic regions was 2.25, lower than the rate at the whole genome level (Supplementary Table 11). We also observed a higher than expected occurrence of very-closely clustered mutations ($P = 2.3 \times 10^{-16}$), which is a common phenomenon in many organisms[44, 46]. Multiple mechanisms, including compound mutation and adaptive natural selection, have been proposed to explain this phenomenon, but none have been directly confirmed[38].

## Discussion

The sequencing and assembly of large, highly repetitive, and heterozygous genomes rich in copy number variants (CNVs) is extremely challenging. In this study, we de novo assembled one maize and one *mexicana* genome based on a new genetic design (Fig. 1) that eliminates assembly problems associated with the heterozygosity of the *mexicana* parent. This design divided the two genomes into 211 and 176 unique chromosomal segments originating from selected individuals of a maize Mo17 × *mexicana*

**Fig. 3** The relationship between *Inv9d* and phenotype. **a** LD reveals the large inversion in *Zea mays. mexicana* based on 56 K SNPs in 95 *mexicana* accessions. LD ($r^2 \geq 0.1$, red; $r^2 <0.1$, pink; $P < 0.1$, black; $P \geq 0.1$, gray) for pairs of SNPs are shown across chromosome 9. Dashed black lines delineate the likely boundaries of structural variants discussed in the text. **b** QTLs for ear-leaf width located in *Inv9d* have been detected in 6 environments. The dashed line indicates the cutoff of LOD value (LOD ≥ 2.5). **c** The dot-plot of gene co-linearity between B73 and *mexicana* on chromosome 9. The histogram indicated the gene density in a 1 Mb window. The green block covered the gene density in the *Inv9d* region. **d** The gene co-linearity between chromosome 9 of B73 and chromosome 6 of rice. **e** The gene co-linearity between chromosome 9 of *mexicana* and chromosome 6 of rice. **f** The phenotype of ear-leaf width of the 10 sequenced lines. The phenotype of *mexicana* haplotype with *Inv9d* (orange) and Mo17 haplotypes without *Inv9d* (green) are shown. The length of scale bar is 10 cm

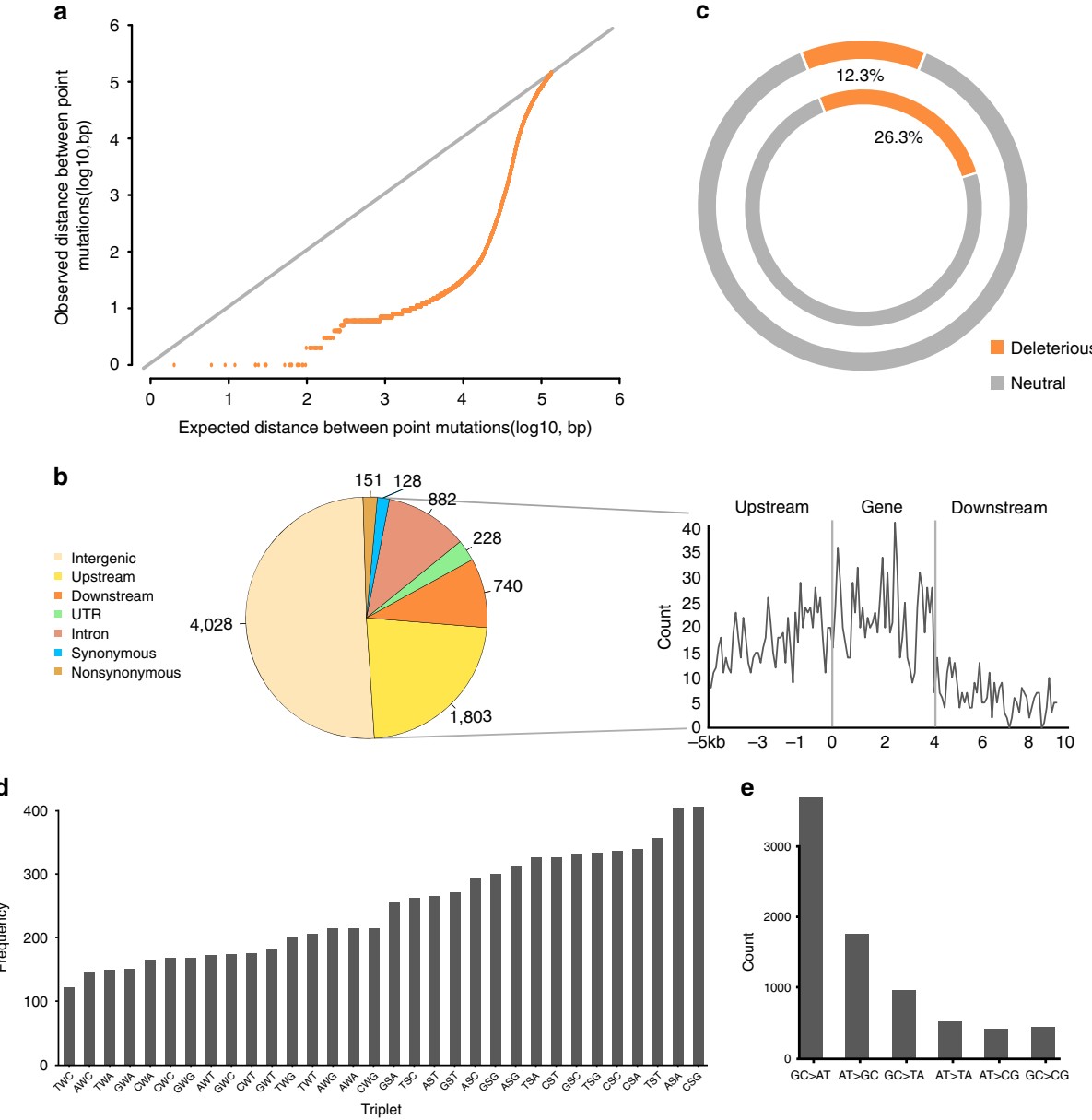

**Fig. 4** The characteristics of spontaneous mutations. **a** The distribution of mutations was not random. The distances between adjacent mutations (observed, orange) against a random distribution of mutation across the genome (expected, gray). **b** The proportion of deleterious mutations in centromeres (the inner circle) and chromosome arms (the outer circle) are shown in orange. **c** The distribution of mutations by functional classification. The relative positions of mutations in gene regions were normalized referring to an average length (4 kb) based on B73 gene models[4]. **d** Variation in mutation rate of different triplets. **e** The mutation number of transitions and transversions

segregating population, derived from a single $F_1$ individual. The meta-assembly approach included high-coverage Illumina, DenovoMAGIC 2™ sequencing, low-coverage PacBio sequencing, and a high-density linkage map of the TM population. The contribution of each method is shown (Fig. 1c). However, the genetic design of the $BC_2F_7$ population made the sequencing depth of the Mo17 genome about 7 times higher, which resulted in lower coverage of the *mexicana* genome. Double haploid or other balanced populations could be used in the future to overcome this. The present Mo17 and *mexicana* draft genomes are considered "version 1.0", due to the technical limitations of short read assembly and *Zea* genome complexity[4] and could be improved by using additional long-read assembly strategies[47]. Through comparative genomic analysis, 220,860 (~ 88.8 Mb PAVs between B73 and Mo17 including 1293 PAV genes were

identified. However, given the sequence bias of Illumina and the low coverage of PacBio, the number of PAVs might be slightly overestimated.

Our genetic design provided an opportunity to study spontaneous mutation rates at a genome-wide level. In the present study, the mutation rate was $2.17 \times 10^{-8} \sim 3.87 \times 10^{-8}$ per site per generation, slightly higher than in other species[37, 48], which may contribute to the abundant diversity that exists in the maize genome[49]. Our estimate is also higher than a previous one based on IBD analysis from maize inbreds[35], possibly due to the high overall heterozygosity and divergent parents in the early TM population generations[49]. A higher deleterious mutation rate was observed in pericentromeric regions, where recombination was suppressed, and helps to explain the excess residual heterozygosity in pericentromeric regions[50], since heterozygosity

increases tolerance to harmful mutations. Transcription-related processes could explain the higher mutation rate in genic regions[51]. When DNA is single-stranded state, there is no protection of nucleosomes, therefore, it might be mutated more easily[52]. A nonrandom distribution of mutations was also observed outside of the centromere regions similar to other species. The underlying mechanisms are still unclear, though a number of hypotheses have been proposed[46, 49, 53]. A high-resolution recombination maps constructed on the basis of single-microspore sequencing[54] will help in the study of the relationships between mutation and recombination[38, 55] and of gene conversion[56].

We estimated that about 10.7% of maize genomic regions were introgressed from *mexicana*, however, the size and number of introgressed regions for each individual maize line may have been underestimated due to limited marker density. Introgression can be associated with maize adaptation from lowland to highland environments, and may have contributed to genetic improvement by increasing yield potential and other important agronomical traits. The TM population provided a unique resource to identify a number of introgression regions that potentially affect phenotypes and important agronomic traits that were supported by previous and present QTL studies. Because limited *mexicana* genomic resources were available, the QTL detection power and the phenotypic effects of the introgression regions may be underestimated. The developed TM population provided a good resource to identify the underlying genes for introgression and other important agronomic traits derived from the *mexicana* genome that had been largely ignored in the past. Future studies will enhance our understanding of introgression, adaptation, and breeding.

## Methods

**Plant material and DNA sequencing**. The $BC_2F_7$ population (containing 191 individuals) was derived from a single $F_1$ seed from a cross between Mo17 and *Zea. mays. ssp. mexicana* (PI 566673); each individual theoretically containing 12.5% (actually, 11.22%) *mexicana* and 87.5% (actually, 88.78%) Mo17 genome fragments. Ten individuals (Fig. 1a) derived from the TM population were chosen for genome assembly, to cover as much as possible of *mexicana* genome, based on the high-resolution linkage map. These individuals contain DNA segments covering the whole Mo17 genome and approximately 96% *mexicana* genome (16.48% on average for each selected individual). We assigned the assemblies of the two parental genomes into separate bins (Fig. 1a and Supplementary Data 1). Young leaves in each individual were used for paired-end-tag DNA sequencing using Illumina HiSeq2000 platform. Millions of reads were generated from libraries with different fragment sizes (175 bp, 300 bp, 500 bp, 3 kb, 9 kb, and 12 kb, Supplementary Table 1). Moreover, Mo17, TM24, and TM104 were sequenced on a Pacific Biosciences (PacBio) platform with 10 Single Molecule Real-Time (SMRT) cells (Supplementary Table 2), and TM104 was further sequenced and assembled using DenovoMAGIC 2^TM (http://nrgene.com/products-technology/denovomagic/) (Supplementary Table 3). See the Supplementary Note 4 for additional details.

**Genome assembly with a meta-assembly pipeline**. A meta-assembly pipeline was developed to assemble Mo17 and *mexicana* genomes (Fig. 1c). We filtered the Illumina reads and removed adapters for each library. The assembly can be divided into three major steps: Step 1, three strategies were executed to assemble contigs, including de novo assembly of 10 individuals, reference-based assembly based on B73 genome, and de novo assembly of unmapped reads. Each of the 10 individuals was de novo assembled with SOAPdenovo2[57] (V2.04). Reads of all libraries were aligned to the maize B73 reference genome (RefGen_v2 release 5b)[4], the mapped reads were assembled with MaSuRCA[58] (V.2.1.0) in each bin, and unmapped reads were de novo assembled with MaSuRCA[58] (V.2.1.0); Step 2, the contigs of the above three strategies were merged, and were further connected or extended with Pacbio long reads (Supplementary Fig. 1, Supplementary Note 1). The PacBio long reads were filtered and aligned to B73 genome[4] by using BLASR[59]. After integrating the contigs, long mate-pair libraries were added to assemble scaffolds with SSPACE[60]. SOAP GapCloser[57] (V1.12), GapFiller[61], and Pbjelly[62] were used to bridge scaffold gaps with paired-end reads and PacBio long reads to obtain scaffolds; and Step 3, the NRGene scaffolds were integrated, and the merged scaffolds were the final scaffolds (Supplementary Note 1).

**Anchoring of the assembled scaffolds to genetic map**. A high-density genetic linkage map was developed using the TM population with 191 recombinant inbred lines and genotyped with the Illumina MaizeSNP50 array[12]. In addition, the 10 sequenced individuals were genotyped using a maize 600k Affymetrix SNP array[63]. First, scaffolds were ranked using B73 reference position by aligning the probes of the 600k SNP array to scaffolds. Second, the genetic linkage maps of TM population and integrated B73 and Mo17 (IBM) population were used to adjust structure variations and mis-assembly of Mo17 and *mexicana* genomes. Finally five scaffolds of Mo17 from NRGene were corrected by comparison with the genetic and physical maps (Supplementary Table 12). The remaining scaffolds were anchored using genotype by sequencing (GBS) probes[6].

**Identification of repetitive elements**. Transposable element (TE) libraries were constructed with RepeatModeler[64], and were applied to mask the Mo17 and *mexicana* genomes by using RepeatMasker software (http://www.repeatmasker.org/).

**Gene prediction and annotation**. An integrated approach combining de novo prediction with evidence-based data (ESTs, protein homology, and RNA-seq) analysis was employed by using the PASA[65] and EVM[66] pipeline. The predicted gene models from EVM[66] were then updated by PASA[65] assembly alignments. Gene functions were assigned according to the best alignment using BLASTP[67] ($E$ value < $10^{-5}$) to the UniProt database[68]. InterProScan[69] was used to identify gene ontology terms, motifs, and domains of gene models. See the Supplementary Note 3 for additional details.

**PAV detection**. Two approaches were employed to identify PAVs between B73 and Mo17, (Supplementary Fig. 5a). The sequences uniquely present in Mo17 compared with the B73 genome were identified using approach 1, consisting of the following steps: (1) Mo17 scaffolds were aligned to the B73 genome with NUC-mer[70] (-c 90 -l 40–maxmatch); (2) Unaligned sequences in Mo17 were extracted and gap regions containing undetermined N bases were filtered out; (3) The filtered Mo17 sequences (>100 bp) were aligned to the B73 genome with BLASTN[67] (-evalue 1e-5 -perc_identity 90); and (4) Unaligned sequences were kept as variations uniquely present in Mo17.

The sequences uniquely present in B73 as compared with Mo17 were identified using approach 2, consisting of the following steps: (1) Mo17 reads were mapped to the B73 genome with bwa[71] (V0.7.4); (2) Regions (≤50 bp) with coverage depth ≤ 2 were merged; (3) Genomic sequences in the merged regions were extracted and aligned to the other genomes with BLASTN[67]; and (4) Unaligned regions (>100 bp) without N bases were kept as the unique PAVs.

**SNP calling and cross validation**. Sequencing reads from 10 individuals of the TM population were mapped to the maize B73 reference genome (RefGen_v2 release 5b)[4] using bwa[71] (V0.7.4) with default parameters. Only reads mapping to unique sites were retained for SNP calling. We used the rmdup function in SAMtools[72] to filter PCR duplicates. SNPs were first called using UnifiedGenotyper of GATK[73] (V 2.7–2), the results were used as the input files to the local realignment tool to minimize the number of mismatch bases. The calmd function of SAMtools[72] was used to improve SNP specificity. Finally, SNPs were called using UnifiedGenotyper of GATK[73] and mpileup of SAMtools[72]. The final SNPs were filtered as follows: (1) the low mapping quality SNPs (MQ < 30) were removed; (2) the read depths of high-quality SNPs were in the range of 5 ~ 40; (3) the distance of two adjacent SNPs was ≥ 5 bp; and (4) the depth of minor allele was ≥ 5.

In order to verify the reliability of SNPs, the commercially available maize 600k Affymetrix SNP array[63] was used to identify SNPs of the 10 selected TM lines. The overall mean concordance rate was 99.5%, and the mean SNPs heterozygous rate was lower than 5% (Supplementary Table 13).

**Mutation identification**. The 10 sequenced individuals from $BC_2F_7$ population were derived from a single $F_1$ individual and covered Mo17 genomes more than 8 times, which provided the opportunity to detect the spontaneous mutations during the 10 continuous generation sections by using the shared Mo17 IBD regions. A preliminary candidate mutation was called if one individual had a different allele compared with other individuals in the Mo17 IBD regions. Rigorous filtering criteria were employed to ensure the accuracy of the mutation identification: the candidate mutation must be supported by at least 5 uniquely mapped high-quality reads (MQ ≥ 50 for each of the two alleles & the paired reads were also well mapped onto the same chromosome). Excluded from analysis were those regions with unexpected and clustered SNPs identified that could be caused by residual heterozygosity of Mo17, which was used three times and from different individuals each time for producing the TM population (Fig. 1a). In total, nearly 2000 mutations were detected in two or more individuals in the 10 sequenced lines (Supplementary Fig. 9), which were also excluded from analysis. This might be due to the preexistent mutations in the parent of Mo17, as mentioned above.

**Introgression analysis**. We combined the data of the present two genomes and public genomic data (including 2 *mexicana*, 18 *parviglumis* and 895 maize[31]) to

detect the introgression regions by using the population approach[74]. First, combining our SNPs of *mexicana* with the SNPs of teosintes obtained from Hapmap3[39], the IBD regions of maize to the *mexicana* and *parviglumis* subpopulation were identified via fastIBD[75]. The genome was divided into 100 kb size windows and the number of recorded IBD tracts between maize and the *mexicana* and *parviglumis* subpopulations was calculated in each window, respectively. The calculated numbers were normalized from 0 (no IBD detected) to 1 (IBD shared by all individuals within a subpopulation), considering that the total number of pairwise comparisons differed between the subpopulations. The normalized IBD between maize and *mexicana* subpopulation ($nIBD_{mex}$) and the normalized IBD between maize and *parviglumis* subpopulation ($nIBD_{par}$) were then used to calculate the relative IBD between the compared groups ($rIBD = nIBD_{mex} - nIBD_{par}$). The detection of *mexicana* introgression was based on patterns of allele sharing and IBD, which may be biased toward low diversity regions. The bins whose SNPs density was lower than the average were filtered. A bin with $\frac{rIBD - \mu}{\sigma \times rIBD} > 2$ was regarded as a putative introgression region (Supplementary Fig. 6a), $\mu$ indicated the mean of $rIBD$ and $\sigma$ indicated the standard deviation of $rIBD$. However, we used 500 kb-sized window to detect the introgression regions in landraces because of the low SNP density (SNP number: 946,072).

**Phenotyping and QTL mapping**. Field trials of TM population (191 individuals) were conducted in 10 environments. Each line was grown in a single 3 m row with planting density of 45,000 plants/ha. Five randomly selected plants in each line were used for measuring ear-leaf width and kernel row number. The BLUP (best linear unbiased prediction) values for both traits were used for QTL mapping.

All the TM lines and their parents were genotyped using the Illumina MaizeSNP50 array[12]. To construct the genetic linkage map, a method was developed integrating the Carthagene software[76] in a Linux system with in-house Perl scripts (https://github.com/panqingchun/linkage_map). Markers that completely co-localized were assigned into a chromosomal bin. Each bin was considered one marker. 12,390 polymorphic markers were obtained in total, which were incorporated into 1,282 unique bins (markers) in this study[77]. QTL mapping using the composite interval mapping method[78] was performed in the package QTL Cartographer version 2.5[79].

**Data availability**. The Mo17 and *mexicana* genome have been submitted to GenBank under the accession codes LMUZ00000000 and LMVA00000000. A website to access the data has also been made available at http://mmgdb.hzau.edu.cn/maize/. The authors declare that all other data supporting the findings of this study are available from the corresponding authors upon request.

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

## Acknowledgements

We greatly appreciate Prof. Jiansheng Li at China Agricultural University for significant contribution in developing the TM population and Drs Edward Buckler and Fei Lu at Cornell University for evaluating the assemble quality using GBS anchors. This research was supported by the National Key Research and Development Program of China (2016YFD0101003), the National Natural Science Foundation of China (31525017 and 31571351, 91435205), the National Youth Top-notch Talent Support Program, and the Fundamental Research Funds for the Central Universities.

## Author contributions

J.Y. and L-L.C.: designed and supervised this study. Li.C. and X.Y.: developed the materials. N.Y., X.-W.X., R.-R.W., W.-L.P., J.-M.S., X.L., L.N., L.C., J.L. L.W., and D.J.: performed the data analysis. Y.W., M.J., and M.D.: conducted the PCR validation experiments. W.L.: managed the field work. Q.P. and F.L.: constructed the linkage map. N.Y., X.-W.X., L.-L.C., and J.Y.: prepared the manuscript. D.J. and X.Y.: helped to edit the manuscript. All the authors read and approved the manuscript.

## Additional information

**Competing interests:** The authors declare thet they have no competing financial interests.

