## [Peer Review File · Nature Communications]

Reviewers' comments:

Reviewer #1 (Remarks to the Author):

This paper reports the sequencing of a MO17 maize genome along with a genome from *Z. mays mexicana*. To these extent that these are good genomes, their addition to the public sphere would be an impressive contribution. To generate these genomes, the authors use a unique back-crossing approach, coupled with anchoring the physical genome to the backcrossed population. I understand their approach conceptually, and it seems in the end to provide two reasonable genomes. I will confess, though, that I find it difficult to evaluate the genomes carefully within the context of this short paper. [Because of their approach, there is a non-trivial issue assigning sequences to one of the two parents.] They do provide a number of statistics, such as BUSCO analyses, which suggest the genomes are relatively complete. It'd be nice to have N50s, too, but overall I'm willing to suspect disbelief and take the genomes at face-value as important contributions to the literature.

Given these genomes, the authors perform a series of standard analyses, such as identification of the number of genes and the number of gene families shared among genomes. The comments in lines 87 to 90 confused me... how does the # of conserved gene families inform us about the pan-genome? I'm not sure the conclusion follows from the data.

lines 110 and following - the identification of structural variation is important and somewhat difficult. These genomes, which are enriched by PacBio reads, should provide some basis for insight. While I enjoyed all of the observations about structural variants, I would love to have just a few sentences about how SVs were inferred. I realize that this information is in the Materials, but without some hint of methods it is difficult to evaluate the results.

line 116 - What is a TE-related PAV? Is this the loss of a full gene, not just a fragment, with deletion endpoints that map to TEs? Or are these TEs that differ in presence and absence between accessions? What is the average size of these PAVs?

The three PAVs > 1 Mb are really neat. line 149 "is caused by changes..." might be better as "may be caused by changes...". I didn't understand line 151 and following - i.e., the sentence beginning "The gradient of gene density..."

line 172 - it would be appropriate to say something about how 10.7% was inferred. I'm not sure how this relates to introgression regions of 0.005% or what these regions actually are. Something is not adding up here.

line 186 - three out of how many inferred QTL?

I don't see much value in the paragraph in lines 188 to 206. I'm assuming that these tests were applied with a single assumed phylogeny. If that phylogeny doesn't hold, due to wide-spread lineage sorting and/or introgression, then the inferred positively-selected genes are likely to be false positives. If phylogeny and positive selection were estimated separately for different genes, then the approach might be more valuable. Still, the *fdr* must be very high, and I have no confidence in the results. I'd just delete the paragraph - there are plenty of other interesting things in the paper.

The estimate of point mutation rate is neat. Presumably only a small subset of the genome was comparable across the 10 TM individuals for Mo17 segments. Can the length of the comparable regions be reported, along with the number of raw observations (e.g., number of conserved and variable sites)

How is hypermethylation related to recombination (line 233). Sorry, but I don't follow this sentence at all.

line 245 - context dependency has been measured in maize - Morton et al., 2006 Genetics

line 240 - several other plants have shown this pericentromeric effect - see Kono et al., MBE 2016; Liu et al MBE 2017

line, 269 - so just what is the coverage and N50 of the mexicana genome? These should be reported.

line 280 - reference 47 may be wrong here?

line 282 - it is actually a higher RATE of deleterious mutation (per base pair) or an enrichment of deleterious mutations (relative to synonymous mutations). This is not clear to me from lines 240.

Reviewer #2 (Remarks to the Author):

In their publication, Yang et al. present two novel maize genomes, one for Mo17, a modern variety, and one for a wild relative (teosinte). They present a new method of generating a high quality assembly of the two varieties – they sequenced 10 individuals from a F1 cross population between Mo17 and teosinte to find regions that genetically divided both parental lines by aligning the reads with the B73 reference and dividing mapping and unmapping reads to assemble both groups, then additionally sequenced two individuals and Mo17 using PacBio, and furthermore sequenced one individual using NRGene DenovoMAGIC, and merged these assemblies into two assemblies. This is certainly novel but I cannot see a problem with this approach, it should certainly fix most of the errors associated with PacBio and nowadays NRGene's approach.

From there the paper becomes a bit more conventional, but still very comprehensive – anchoring via genetic maps, repeatmasking, annotation, SNP calling, IBD mapping, checking for (positive) selection, inversions and PAV. They identify some large PAVs that they cannot link to phenotypes, so more phenotyping could improve this study based on the annotation of the genes in the lost regions, but it could quickly become a fishing expedition.

I have one problem with this sentence: ` These results indicated that the mexicana genome was the ancestor state, and that the modern maize genome was derived from the inversion. ` This is based on the fact that mexicana genome doesn't carry an inversion that the modern maize genome carries, an inversion that also doesn't appear in rice in the same syntenic block. The formulation is just a bit sloppy and could be interpreted that the mexicana genome is the ancestral genome, but in reality it means (I think) that mexicana is closer to the ancestral (probably extinct) genome than Mo73, even though this inversion seems to appear in some accessions ('this inversion was only found in some mexicana accessions, and not in parviglumis in a previous studies')? Anyway, I'd rewrite that sentence to clarify that mexicana doesn't have to be the ancestral genome.

My biggest problem is that there is a serious lack of biology in this paper, there is a strong focus on bioinformatics/population genetics – there is so much more that could be done, for example, with the list of positively selected genes. Right now the paper lists a few names and their rough role but stops there. Open questions remain – in ZMex05g017761, how do the 4 positively selected sites affect the function? ZMex05g020691 is a member of the bZIP transcription factor family, what does it interact with, why could it be under positive selection? In which pathways are genes under selection located? I

assume many of the nonsynonymous SNPs introduced stop codons leading to probably pseudogenes, what were these genes involved in? I'm also surprised that absolutely nothing related to plant breeding appeared in these genes under selection in B73 and Mo17 - perhaps because the authors looked only at single copy genes, perhaps the authors should also have looked at genes undergoing negative selection. A stronger focus on the history of plant breeding as it is written in these three genomes would make this a very strong paper, I think that the authors have most of the data necessary to find that story.

I cannot see any large faults with the analyses performed, but I would really like to see more biology to warrant publication

Reviewer #3 (Remarks to the Author):

Review of the paper Yang et al. "contributions of teosinte haplotypes to modern maize".

The article presents an extensive resequencing and assembly effort to study maize and teosinte diversity, the occurrence of introgression and mutation pattern in maize. The article assembled the genome of an inbred MO17 and a Mexicana genome. The authors presented thorough analyses, however, there is some issues with the result of some of these analyses that need to be addressed and can have direct impact on the main conclusions. At this stage I recommend a rejection, but indicate to the authors that further work might justify a resubmission.

Assembly

One major issue is the size difference between B73/MO17 and the Mexicana genome. So B73 is roughly 2.3G, MO17 in this study 2.04G and Mexicana 1.2G. Variation of genome sizes between maize varieties around 30% are common, but the size of the Mexicana genome seems much smaller only 50% of B73. It seems quite extreme. The author quality assessment show that at the gene level 86% of BUSCO genes are mapped and only 81.5% of B73 genes mapped to Mexicana (versus 98.2% for MO17). The gene set is the easiest to be assembled (lower complexity, low redundancy, ...) and show quite a significance difference. Certainly a large fraction in the repetitive part of the genome is not assembled and we certainly have also a partial assembly of the gene set for Mexicana. We have 31 387 gene models for Mexicana versus 40003 for MO17 and 39324 for the v4 version of B73. So very similar between MO17/B73 but 80% of B73 for Mexicana (roughly the estimation made by the author).

L105. The conclusion should be slightly different. Certainly MO17 for genic regions is rather good, less so for Mexicana. The genome size is drastically smaller, an unexpected and certainly a consequence of not very well assembled repetitive region. It is also clear from the contig N50/scaffold N50. Mexicana have only 100kb versus almost 3M.

The consequence of this difference is not clear to me when one consider PAV. With such high difference in assembly, how confident are we in calling/assessing PAV?

Fig S4 show clearly lower number of PAV (count per category of size, whatever the category)

L91. BUSCO gene set. So the article referenced mainly focus on vertebrates, anthropods, metazoan, fungi and broadly eukaryote. So there is not per see a plant reference but the author might speak of eukaryote. Please clarify.

Introgression estimation.

The introgression study is not enough clear. The author states L172 "we found as much as 10.7% ...introgression from Mexicana". So I guess the author speak of all individual together. Please specify. The distribution of the statistic $rIBD$ is not shown to assess its distribution. Neither of the centred reduced version of this statistic $(rIBD - \mu) / \sigma$. A figure with the individual distribution per maize individual might be also provided with the raw data of table S11.

Does maize from highland Mexico have significantly higher proportion? It is an expected results, but do they found it? It will help build their case.

As a control, using parviglumis to assess Mexicana introgression might be a good idea. We expect low introgression but then it will help to assess the %. Moreover, one aspect the author has not considered as a confounding effect is the introgression of maize into Mexicana. In that specific case the introgressed maize fragments into Mexicana (considered Mexicana) might show up as an introgression of Mexicana into Maize. The introgression hotspot of chromosome 8? Please check and prove me otherwise, or allow to acknowledge the limit of the approach.

L185. It is unclear for me what table 11.2 represent. My understanding is 1) introgressed individual individual delimitation from the analysis of all the maize, 2) the QTL performed on the current mapping population. So the delimitation of the analysis is not the same 10kb for introgression and if I understand clearly 1283 bin for QTL analysis with a genome size of 2G it end up around 1.5M bin in average. The result in table 11 is presented with 100kb bin so it is unclear how this table was built.

With very large bin like this, it is likely to found introgression by chance if 10% of the genome is introgressed. A better analysis is to assess if highland maize is introgressed not all the available sample of maize. Again we expect highland maize from Mexica to have higher introgression. So restrict your comparisons of introgression to highland maize from Mexico.

Mutation analysis.

Some confounding effect is the diversity found inside MO17. Any remaining diversity found inside MO17 because of the backcross will then might be call as new mutations. The author say that a single F1 was backcross with MO17 twice (I call MO17-1, the first cross leading to F1, Mo17-2 the first backcross, Mo17-3 the second backcross). We could calculated the probability to have 1 plant only of the 10 plants have the last backcross fragment (MO17-3) , it is 0.0097. If we considered the MO17-2, the probability is 0.187, if we considered MO17-1 then it is 0.376. What does it tell us. That if there is some remaining heterozygosity or initial accumulated mutation in the three parental MO17 and the MO17 are slightly different (MO17-1, MO17-2, MO17-3) then SNP with the procedure use by the author might be considered mutation but will be initial heterozygosity or mutations. So there is certainly an overestimation of the mutation rate using the procedure the author used. The author procedure does not allow to completely correcting it. Moreover, the mutation found here also accumulated in the parental inbred if they were "perfect".

L255. If clustered mutation might not be uncommon, it also reflects remaining heterozygosity.

L219-237. So the mutation rate is estimate through the lence of mapping. Is there difference of mapping quality/depth/uniqueness between gene and intergenic? Does the higher mutation in gene reflect mapping differential?

Reviewer #4 (Remarks to the Author):

In this article, the authors assemble the draft genomes of the Mo17 variety of maize and the teosinte. On the bases of the two assemblies, they identify and characterize several presence-absence variants. As part of the process, they genotype a set of recombinant inbred lines. With this information, they confirm a polymorphic chromosomal inversion on chromosome 9, they search for QTL for some morphological traits, and they also estimate the spontaneous mutation rate during the 10 generations of inbreeding. This is quite an impressive amount of work, and some interesting results, that could become a beautiful publication if some weak points were addressed before. In what follows, I will make some general recommendations, and more detailed ones on the topic of the mutation rate.

The main problem I see with the manuscript is a formal one. It is not well structured, what makes it very difficult to understand.

The introduction should include all the references to the state of the art that are currently cited along the results. Otherwise, it is sometimes difficult to distinguish the original contributions from previously published results. The introduction should also set the goals and motivate the methods. Unfortunately, no justification is given for the most important and original method used: the meta-assembly pipeline. Only in the Discussion we can read that the new genetic design "eliminates assembly problems associated with the heterozygosity of the mexicana parent". However, an evident and simpler approach would have been to inbreed the teosinte and sequence the two genomes separately. Any reader would wonder why the authors mixed up two genomes if they wanted to assemble them separately.

The second problem is that some methods are hardly described. This stems from the excess of results reported. A sharper focus on a few interesting results would improve this aspect, as well as the previous one. For example, the anchoring and ordering of scaffolds with the genetic map of the TM population is not described in the Online methods section. Another example: I did not find an explanation of how the mutations are classified as 'deleterious' or not.

My third main concern is the adequacy of the methods. First, the meta-assembly pipeline does not seem to be the best option to assemble genomes characterized by high structural variation. Even the linkage maps can be tricked by structural variants. The very definition of the genomic bins that are assigned to one parental genome or the other in each TM individual sequenced is based on the coordinates of the B73 genome. The fact that two reads of the same TM individual map to the same bin certainly indicate that they come from the same parental genome, but it does not tell us much about the positions in the parental genome from which they come, unless we assume co-linearity between the genomes of B73, Mo17 and mexicana. It is not clear to me to what extent the assemblies depend on this assumption. Plus, the long reads coming from a TM individual help assemble the genome of that individual, which may differ structurally from any of the parentals. In summary, I need an explanation of how the structural variation between Mo17 and teosinte would affect the meta-assembly approach. This is critical, since the validity of the PAVs identified depends on it. To put it another way, despite the efforts to validate the assemblies, I am not convinced that the quality of the assembly is enough to call PAVs.

The introgression analysis is also questionable. It is a very arbitrary heuristic that should be better justified, or changed to match the state of the art.

My doubts about the adequacy of the methods bring me to the topic of the mutation rate. The ten generations of selfing of the backcrossed population allowed for the accumulation of mutations, that the authors identify in the Mo17 regions as variants represented in a single individual, either heterozygous or homozygous. The authors report a higher mutation rate than previous estimates,

despite having filtered out genomic regions suspected to contain ancestral heterozygosity. The authors are aware that the genomic fragments of Mo17 ancestry in the recombinant inbred lines do not come all from the same individual, but from three of them. Given the high estimate, it is reasonable to wonder if the mutation rate is biased by existing variation among the three Mo17 genomes introduced in these lines, despite of the measures taken to prevent this bias. I suggest one way to validate their estimate.

The genomic fragments with mexicana ancestry do come all from the same genome copy in a single individual, and they are expected to experience the same mutation rate than the Mo17 fragments along the 10 generations of inbreeding. Despite a lower coverage, that prevents an accurate estimate of the mutation rate from the mexicana fragments, it should be possible to compare the number of mutations observed in the mexicana fragments that are covered in two or more TM individuals with the expectation under the mutation rate estimated in the Mo17 fragments.

J. Ignacio Lucas Lledó

Reviewers' comments:

Reviewer #1 (Remarks to the Author):

This paper reports the sequencing of a MO17 maize genome along with a genome from *Z. mays mexicana*. To these extent that these are good genomes, there addition to the public sphere would be an impressive contribution. To generate these genomes, the authors use a unique back-crossing approach, coupled with anchoring the physical genome to the backcrossed population. I understand their approach conceptually, and it seems in the end to provide two reasonable genomes. I will confess, though, that I find it difficult to evaluate the genomes carefully within the context of this short paper. [Because of their approach, there is a non-trivial issue assigning sequences to one of the two parents.]

Q1: They do provide a number of statistics, such as BUSCO analyses, which suggest the genomes are relatively complete. It'd be nice to have N50s, too, but overall I'm willing to suspect disbelief and take the genomes at face-value as important contributions to the literature.

[Response]: We have described the assemble strategy in detail in the manuscript. We are quite confident with the strategy and results. To further validate the genome assembly, we have sequenced newly four Mo17 BACs using PacBio platform (with one cell) and compared them with the assembled genome of Mo17. It is observed that these BAC sequences showed high sequence identity with our Mo17 assembled genome (see following figure), indicating that the assembled quality is reliable.

Fig. Comparison of assembled genome and sequenced BACs

Q2: Given these genomes, the authors perform a series of standard analyses, such as identification of the number of genes and the number of gene families shared among genomes. The comments in lines 87 to 90 confused me... how does the # of conserved gene families inform us about the pan-genome? I'm not sure the conclusion follows from the data.

[Response]: Sorry for the confusing description. Considering that only 2,000 conserved protein families were identified among B73, Mo17, and *mexicana* genomes, which is similar to the comparison of B73 with other grass species such as rice and sorghum. We indicate that different *Zea* genomes have high genetic diversity, therefore it is highly important to construct a pan-genome of *Zea mays*. To avoid misleading, we delete this sentence.

Q3: lines 110 and following - the identification of structural variation is important and somewhat difficult. These genomes, which are enriched by PacBio reads, should provide some basis for insight. While I enjoyed all of the observations about structural variants, I would love to have just a few sentences about how SVs were inferred. I realize that this information is in the Materials, but without some hint of methods it is difficult to evaluate the results.

[Response]: Thanks for the reviewer for the constructive comments. The detailed process to identify SVs was listed in online methods and Supplementary Materials. Several sentences describing the identification of SVs have been added to the Result section.

Q4: line 116 - What is a TE-related PAV? Is this the loss of a full gene, not just a fragment, with deletion endpoints that map to TEs? Or are these TEs that differ in presence and absence between accessions? What is the average size of these PAVs?

[Response]: We used Hmmer tool to search the TE domain on the specific sequences of B73, Mo17 and *mexicana*, if the specific sequences contained TE domain and the TE content $\geq 80\%$, it is regarded as a TE-related PAV (Zhang *et al.*, *Plant Cell*. 27,1595-1604,2015). The average size of these PAV was shown in the following table:

Classification	Average length
B73&Mo17	713 bp
B73& mexicana	698 bp
Mo17& mexicana	518 bp

Considering that the *mexicana* genome is incomplete, we only kept the presence/absence variations (PAVs, ≥ 100 bp) between B73 and Mo17 in the revised manuscript.

Q5: The three PAVs > 1 Mb are really neat. line 149 "is caused by changes..." might be better as "may be caused by changes...". I didn't understand line 151 and following - i.e., the sentence beginning "The gradient of gene density..."

[Response]: Thank you for the praise and suggestions. We have revised it to be "may be caused by changes...". In terms of the sentence beginning "The gradient of gene density...", this sentence just demonstrated two facts to prove the *mexicana* genome in *Inv9d* region should be closer to the ancestor state. Generally, the gradient of gene density should be increased towards chromosome ends, but this common gradient was disturbed in the B73 homologous region of *Inv9d*. And the shorter length of B73 homologous region of *Inv9d* indicated the possible changes in rates of DNA loss and gain in regions that switch from chromosome ends to the near pericentromeric contexts (Bertioli *et al.*, *Nat Genet.* 48, 438-446, 2016).

Q6: line 172 - it would be appropriate to say something about how 10.7% was inferred. I'm not sure how this relates to introgression regions of 0.005% or what these regions actually are. Something is not adding up here.

[Response]: Sorry for the misleading. 10.7% is the sum of unique introgression regions from all individuals in hapmap3. The ratio of introgression region of one single individual is in the range of 0.005% ~ 0.724%. We have added detailed description in the revised manuscript.

Q7: line 186 - three out of how many inferred QTL?

[Response]: Three out of ten inferred QTLs (N. Lauter *et al.*, *Genetics.* 4, 1949-1959, 2004)

Q8: I don't see much value in the paragraph in lines 188 to 206. I'm assuming that

these tests were applied with a single assumed phylogeny. If that phylogeny doesn't hold, due to wide-spread lineage sorting and/or introgression, then the inferred positively-selected genes are likely to be false positives. If phylogeny and positive selection were estimated separately for different genes, then the approach might be more valuable. Still, the *fdr* must be very high, and I have no confidence in the results. I'd just delete the paragraph - there are plenty of other interesting things in the paper.

[Response]: We indeed used an assumed phylogeny as follows: ((Setaria,(Sorghum,(mexicana1,(Mo17,B73))), (Rice,Brachypodium)).

The detailed steps for positive selection analysis were described in the **Supplementary notes**. We used branch-site likelihood ratio method to detect positively selected genes (PSGs). The branch-site model allows us to vary both among sites in the protein and across branches on the tree and aim to detect positive selection affecting a few sites along particular lineages (called foreground branches), therefore we need a phylogenetic tree to separate foreground and background branches. The phylogenetic tree was constructed with 3,887 single copy orthologs and the phylogenetic tree was conformed the existing evolutionary relationship. Although the phylogenetic tree was built with the same single copy orthologs to find PSGs, we only used the phylogenetic tree to separate foreground and background branches, the divergence time and branch length were measured with PAML using the expected number of nucleotide substitutions per codon. All of the 3,887 single copy genes were not located in introgression regions identified in this study, but the introgression region might be underestimated due to limited marker density. If the single copy gene was located in introgression region, which can cause false negative prediction, because B73, Mo17 and *mexicana* were not located in foreground or background branches at the same time, and branch-site likelihood ratio method assumed only the foreground branch may undergo positive selection. We used a relatively strict threshold FDR 0.05 to identify PSGs.

Q9: The estimate of point mutation rate is neat. Presumably only a small subset of the genome was comparable across the 10 TM individuals for Mo17 segments. Can the length of the comparable regions be reported, along with the number of raw observations (e.g., number of conserved and variable sites)

[Response]: We appreciate to see that the reviewer agrees with us in this point. Actually we compared the whole Mo17 genome. When we calculated the point mutation rate we must take the length of compared Mo17 segments into consideration. Because most of genome segments (87.5%) of TM individuals (BC₂F₇) inherited Mo17 genome in theory. In terms of the sequenced 10 individuals, there were at least 5 individuals covering the unique Mo17 bins. So we had the opportunity to evaluate the point mutation rate in the whole genome level.

Q10: How is hypermethylation related to recombination (line 233). Sorry, but I don't follow this sentence at all.

[Response]: Hypermethylation and recombination were both thought as the possible

mechanisms for generating mutations. In *Arabidopsis*, the hypermethylation was inferred to be partially responsible for the higher mutation rate in centromere regions (Ossowski S *et al.*, *Science*. 327: 92-94, 2010). In maize, however, the hypermethylated centromere regions (Regulski M *et al.*, *Genome Res*. 23: 1651-1662, 2013; Palmer *et al.*, *Science*. 302: 2115-2117, 2003) show a lower mutation rate than that in chromosome arms where the recombination rate was promoted. So we thought that the recombination effect might cover up the hypermethylation effect in maize.

Q11: line 245 - context dependency has been measured in maize - Morton et al., 2006 Genetics

[Response]: Thank you for your comments. We have read this article carefully and cited it (Morton *et al.*, *Genetics*. 172: 569-577, 2006). They mentioned that GC->AT pressure and the transition rate increases with increasing locus regional A+T content and GC-> AT pressure decreases with increasing flanking base A+T content, which was consistent with our results (Figure 4). Furthermore, we investigated all the triplet combinations to evaluate the content effect from two immediate neighbors and found that “ASA” and “CSG” (S= “C” or “G”) were more common.

Q12: line 240 - several other plants have shown this pericentromeric effect - see Kono et al., MBE 2016; Liu et al MBE 2017

[Response]: Thank you for your comments. We have read these two valuable articles carefully and found our results are consistent with their conclusions. Kono *et al.* pointed out that the effective recombination rate strongly influences the purging of deleterious variants from populations and found the proportion of nonsynonymous SNPs inferred to be deleterious was higher in pericentromeric region in soybean (Kono *et al.*, *Mol Biol Evol*. 34: 908-924, 2017). Liu *et al.* also found the deleterious variants were enriched within low recombination regions in Asian rice (*O. sativa*) and their wild relatives (*O. rufipogon*). We have added some discussions in the revised manuscript.

Q13: line, 269 - so just what is the coverage and N50 of the mexicana genome? These should be reported.

[Response]: Referring to B73 V2 genome, the *mexicana* bins coverage 96.4% genome (line 62), however, we only assembled about half of the *mexicana* genome, and its contig N50 is 26,638 bp (Table 1).

Q14: line 280 - reference 47 may be wrong here?

[Response]: Thank you for your reminding, we have corrected it in the revised manuscript.

Q15: line 282 - it is actually a higher RATE of deleterious mutation (per base pair) or an enrichment of deleterious mutations (relative to synonymous mutations). This is not clear to me from lines 240.

[Response]: It is actually that deleterious mutations were enriched in protein-coding regions of pericentromeric genes. But not relative to synonymous mutations, deleterious mutations were predicted by PROVEAN (**P**rotein **V**ariation **E**ffect **A**nalyzer), even a nonsynonymous mutation might not be predicted to be a deleterious mutation.

Reviewer #2 (Remarks to the Author):

In their publication, Yang et al. present two novel maize genomes, one for Mo17, a modern variety, and one for a wild relative (teosinte).

They present a new method of generating a high quality assembly of the two varieties – they sequenced 10 individuals from a F1 cross population between Mo17 and teosinte to find regions that genetically divided both parental lines by aligning the reads with the B73 reference and dividing mapping and unmapping reads to assemble both groups, then additionally sequenced two individuals and Mo17 using PacBio, and furthermore sequenced one individual using NRGene DenovoMAGIC, and merged these assemblies into two assemblies. This is certainly novel but I cannot see a problem with this approach, it should certainly fix most of the errors associated with PacBio and nowadays NRGene’s approach.

Q16: From there the paper becomes a bit more conventional, but still very comprehensive – anchoring via genetic maps, repeatmasking, annotation, SNP calling, IBD mapping, checking for (positive) selection, inversions and PAV. They identify some large PAVs that they cannot link to phenotypes, so more phenotyping could improve this study based on the annotation of the genes in the lost regions, but it could quickly become a fishing expedition.

[Response]: Thank you for the reviewer for the encouraging comments. We hope that the Mo17 and *mexicana* genome will helpful for the maize community, and the identified large PAVs can be linked to phenotypes in the near future.

Q17: I have one problem with this sentence: ‘These results indicated that the *mexicana* genome was the ancestor state, and that the modern maize genome was derived from the inversion. ‘This is based on the fact that *mexicana* genome doesn’t carry an inversion that the modern maize genome carries, an inversion that also doesn’t appear in rice in the same syntenic block. The formulation is just a bit sloppy and could be interpreted that the *mexicana* genome is the ancestral genome, but in reality it means (I think) that *mexicana* is closer to the ancestral (probably extinct) genome than Mo73, even though this inversion seems to appear in some accessions (‘this inversion was only found in some *mexicana* accessions, and not in *parviglumis* in a previous studies’)? Anyway, I’d rewrite that sentence to clarify that *mexicana* doesn’t have to be the ancestral genome.

[Response]: Thank you for the reviewer for the constructive comments. We have rewritten the sentence to clarify that the *mexicana* genome was closer to the ancestor

state.

Q18: My biggest problem is that there is a serious lack of biology in this paper, there is a strong focus on bioinformatics/population genetics – there is so much more that could be done, for example, with the list of positively selected genes. Right now the paper lists a few names and their rough role but stops there. Open questions remain – in ZMex05g017761, how do the 4 positively selected sites affect the function? ZMex05g020691 is a member of the bZIP transcription factor family, what does it interact with, why could it be under positive selection? In which pathways are genes under selection located? I assume many of the nonsynonymous SNPs introduced stop codons leading to probably pseudogenes, what were these genes involved in? I’m also surprised that absolutely nothing related to plant breeding appeared in these genes under selection in B73 and Mo17 - perhaps because the authors looked only at single copy genes, perhaps the authors should also have looked at genes undergoing negative selection. A stronger focus on the history of plant breeding as it is written in these three genomes would make this a very strong paper, I think that the authors have most of the data necessary to find that story.

I cannot see any large faults with the analyses performed, but I would really like to see more biology to warrant publication

[Response]: Thank you for the reviewer for the constructive comments. With these suggestions, we have performed more analysis. We have tried to predict the three-dimensional structure of ZMex05g017761, however, there is no suitable templates. Therefore, it is hard to evaluate how the 4 positively selected sites affect the function. For the bZIP transcription factor family member ZMex05g020691, no experimental or predicted protein-protein interactions were identified. We further conduct the GO analysis for the PSGs. Interestingly, PSGs identified in *mexicana* were enriched in jasmonic acid biosynthetic/metabolic process which was an important phytohormone and related to lots of plant stress tolerances (Ahmad P et al., *Front Plant Sci*, 7, 813, 2016).

Supplementary Figure 10. Go enrichment of PSGs. “Rich factor” means that the ratio

of the number of positive select genes and the number of genes have been annotated in background. (a). *mexicana* (b). maize.

Yes, we do see the PSGs were significantly enriched the QTLs detected in TM population for 18 traits (Fisher' exact test, $P\text{-value}=0.2\times 10^{-3}$), especially for yield related traits (Supplementary Table 13). We guess it may associate with the history of plant breeding although additional evidence still required.

We confirmed the ~27 Mb inversion *Inv9d* was real by the linkage map and the *Inv9d* showed altitudinal clines in environmental association analysis. In our studies, a large QTL for ear leaf width was identified in *Inv9d* region indicating the *Inv9d* not only contributed to adaptation but also plant morphology. Through RNA-seq analysis of the 10 selected TM individuals, we observed no different expression pattern in *Inv9d* region. It's also hard to say the *Inv9d* or a gene in *Inv9d* controlled the adaptation and plant morphology, because the rare recombination in this inversion hindered the QTL fine-mapping. However, it has provided a good example that the inversion region may associate with the maize adaptation.

Reviewer #3 (Remarks to the Author):

Review of the paper Yang et al. "contributions of teosinte haplotypes to modern maize".

The article presents an extensive resequencing and assembly effort to study maize and teosinte diversity, the occurrence of introgression and mutation pattern in maize. The article assembled the genome of an inbred MO17 and a Mexicana genome. The authors presented thorough analyses, however, there is some issues with the result of some of these analyses that need to be addressed and can have direct impact on the main conclusions. At this stage I recommend a rejection, but indicate to the authors that further work might justify a resubmission.

Q19: One major issue is the size difference between B73/MO17 and the *mexicana* genome. So B73 is roughly 2.3G, MO17 in this study 2.04G and *mexicana* 1.2G. Variation of genome sizes between maize varieties around 30% are common, but the size of the *mexicana* genome seems much smaller only 50% of B73. It seems quite extreme. The author quality assessment show that at the gene level 86% of BUSCO genes are mapped and only 81.5% of B73 genes mapped to *mexicana* (versus 98.2% for MO17). The gene set is the easiest to be assembled (lower complexity, low redundancy, ...) and show quite a significance difference. Certainly a large fraction in the repetitive part of the genome is not assembled and we certainly have also a partial assembly of the gene set for *mexicana*. We have 31 387 gene models for *mexicana* versus 40003 for MO17 and 39324 for the v4 version of B73. So very similar between MO17/B73 but 80% of B73 for *mexicana* (roughly the estimation made by the author).

[Response]: Thank you for the reviewer for pointing out the problem. Considering that the genetic design of BC₂F₇ population makes the sequencing depth of the Mo17

genome 7 times of *mexicana*, which resulted in lower coverage of the *mexicana* genome. Therefore, only 1.2G *mexicana* genome is assembled. However, 31,387 gene models were identified in *mexicana* (which is about 80% that of B73), indicating that the most of protein-coding regions are relatedly complete, and most of the un-assembled regions are TE-related sequences. We agree the present genomes especially for *mexicana* genome is very draft, however, they will still be very useful for the community especially for the protein-coding regions.

Q20: L105. The conclusion should be slightly different. Certainly MO17 for genic regions is rather good, less so for *mexicana*. The genome size is drastically smaller, an unexpected and certainly a consequence of not very well assembled repetitive region. It is also clear from the contig N50/scaffold N50. *mexicana* have only 100kb versus almost 3M.

[Response]: Different sequencing depth caused by the meta-assembly strategy resulted in the unbalance assembled quality of Mo17 and *mexicana* genomes. To avoid misunderstanding, we have changed the conclusion as: “In summary, these results support the conclusion that the assembled quality of Mo17 is acceptable. However, the assembled quality of *mexicana* is less good than Mo17, only half of *mexicana* genome is assembled as a consequence of not very well assembled repetitive region. But both of genomes are well assembled in protein-coding regions (Supplementary Table 7).”

Q21: The consequence of this difference is not clear to me when one consider PAV. With such high difference in assembly, how confident are we in calling/assessing PAV? Fig S4 show clearly lower number of PAV (count per category of size, whatever the category)

[Response]: Yes, we agree with you that the quality of assembly highly affects the subsequent analysis including PAVs. In the current analysis, we only identify PAVs in the synteny regions. Considering that the assembled *mexicana* genome is much smaller than Mo17 and B73, we only list the PAVs between Mo17 and B73 in the revised manuscript.

Q22: L91. BUSCO gene set. So the article referenced mainly focus on vertebrates, arthropods, metazoan, fungi and broad eukaryote. So there is not per se a plant reference but the author might speak of eukaryote. Please clarify.

[Response]: Indeed, there is not a plant reference in previous BUSCO gene sets. But we had got the Plantae BUSCO dataset through private email communication. BUSCO also has released the plants datasets now: <http://busco.ezlab.org/>. BUSCO referenced paper mainly focused on vertebrates, arthropods, metazoan, fungi and broad eukaryote. BUSCO also provide the dataset for plants on its website (<http://busco.ezlab.org/>).

Introgression estimation.

Q23: The introgression study is not enough clear. The author states L172 “we found as much as 10.7% ...introgression from Mexicana”. So I guess the author speak of all individual together. Please specify.

[Response]: Thank you for your kindly reminding, different individuals have different proportion of introgression regions. 10.7% was the total number of all individual together. We have specified this in the revised manuscript.

Q24: The distribution of the statistic rIBD is not shown to assess its distribution. Neither of the centred reduced version of this statistic $(rIBD-\mu)/\sigma$. A figure with the individual distribution per maize individual might be also provided with the raw data of table S11.

[Response]: We have taken this suggestion and provided a figure.

Supplementary Figure 6. (a) The selected introgression regions based on the statistic rIBD. The dash line indicated the threshold $(\frac{rIBD-\mu}{\sigma*rIBD} > 2)$, the regions on the right were the final candidate introgression regions. (b) the distribution of introgression regions of each individual. (c) The distribution of the line number of introgressed regions.

Q25: Does maize from highland Mexico have significantly higher proportion? It is an expected results, but do they found it? It will help build their case.

[Response]: Thank you for your constructive comments. As data of the hapmap2 or 3 were collected from numbers of colleges and institutions and the living environment information of the germplasms were not offered in their articles, it is difficult to answer your question using this dataset. So we used the other dataset (SNPs number: 946,072) offered by a flowering-time adaptation study in maize landraces (Romero et al., *Nat Genet*, 49: 476-480, 2017; GBS imputed markers at: <http://hdl.handle.net/11529/10035>). 2,900 landraces were detailed with environmental information (passport data can be accessed upon registration at <http://germinate.seedsofdiscovery.org/maize/>). Using the same introgression detection method, we found that maize from highland Mexico indeed have significantly higher introgression proportion than that in low-elevation (Supplementary Figure 7a, $P\text{-value}=0.002$). Furthermore, we found no matter where the maize came from (including Mexico, Honduras, Guatemala, Panama Costa Rica et al.), the introgression proportion was significantly correlated ($P\text{-value}=9.703e-06$) with the elevation (n=482) (Supplementary Figure 7b). This result indicated that some introgression regions should relate to highland adaptation had been kept when maize expand to other places.

Supplementary Figure 7. a. the positive correlation between elevation and introgression proportion. b. the introgression proportion had significantly difference between highland Mexico maize and lowland maize (low-elevation: <1,200 m above sea level; high-elevation: >1,900 m above sea level)

Q26: As a control, using parviglumis to assess *mexicana* introgression might be a good idea. We expect low introgression but then it will help to assess the %. Moreover, one aspect the author has not considered as a confounding effect is the introgression of maize into *mexicana*. In that specific case the introgressed maize fragments into *mexicana* (considered *mexicana*) might show up as an introgression of *mexicana* into Maize. The introgression hotspot of chromosome 8? Please check and prove me otherwise, or allow to acknowledge the limit of the approach.

[Response]: Thank you for your professional comments and agreement with taking

parviglumis as a control. In the terms of introgression direction, we acknowledge the limit of this rIBD approach and add some descriptions in the revised manuscript. The introgression tends to favor crossing in the direction of teosinte to maize. Several factors limit the extent of introgression from maize to teosinte were summarized based on previous studies: a) Three cross-incompatibility loci (*Tcb1*, *Gal*, *Ga2*)-the barriers especially limit the pollination from maize to *mexicana*. (Evans et al., *Theor Appl Genet*, 103: 259-265, 2001; Kermicle et al., *J Hered*, 101: 737-749, 2010; Kermicle et al., *Maydica*, 51: 219-225, 2006) ; b) Teosinte ears produced a mean of 0.2–0.3 seeds per ear when pollinated with maize pollen based on an experimental estimate and between 90% and 95% of the fruitcases produced on teosinte that was fertilized by maize pollen were sterile (Baltazar et al., *Theor Appl Genet*, 110: 519-526, 2005); c) the flowering time differences (Rodriguez et al., *Maydica*, 51: 383-398, 2006), silk longevity was much shorter for teosinte than for maize (approx. 4 days vs. approx. 11 days) (Baltazar et al., *Theor Appl Genet*, 110: 519-526, 2005); (d) teosinte produced more pollen on a per plant basis than the landraces and commercial hybrid maize (Baltazar et al., *Theor Appl Genet*, 110: 519-526, 2005); (e) teosinte frequently produced lateral branches with silks close to a terminal tassel producing pollen (Baltazar et al., *Theor Appl Genet*, 110: 519-526, 2005). Meanwhile, some studies show some evidence for reciprocal introgression between *mexicana* and highland maize landraces (Fukunaga K et al., *Genetics*, 169: 2241-2254, 2005; Hufford et al., *PLoS Genet*, 9: e1003477, 2013). However, the experimental studies support the hypothesis that gene flow and the subsequent introgression of maize genes into teosinte populations most probably results from crosses where teosinte first pollinates maize (Baltazar et al., *Theor Appl Genet*, 110: 519-526, 2005).

Q27: L185. It is unclear for me what table 11.2 represent. My understanding is 1) introgressed individual individual delimitation from the analysis of all the maize, 2) the QTL performed on the current mapping population. So the delimitation of the analysis is not the same 10kb for introgression and if I understand clearly 1283 bin for QTL analysis with a genome size of 2G it end up around 1.5M bin in average. The result in table 11 is presented with 100kb bin so it is unclear how this table was built.

[Response]: The QTL analysis was based on the genetic map of TM population with 1,283 bins. However, introgression analysis was based on 3,532,416 SNPs data merged with our *mexicana* and the hapmap3 SNPs dataset. Then we used 100 kb sized window to detected introgression regions by rIBD method.

Q28: With very large bin like this, it is likely to found introgression by chance if 10% of the genome is introgressed. A better analysis is to assess if highland maize is introgressed not all the available sample of maize. Again we expect highland maize from Mexica to have higher introgression. So restrict your comparisons of introgression to highland maize from Mexico.

[Response]: The introgression analysis was conducted in a 100 kb window not in the large bins (See the response to Q27). And we have checked whether the highland maize have higher introgression or not in Q25.

Mutation analysis.

Q29: Some confounding effect is the diversity found inside MO17. Any remaining diversity found inside MO17 because of the backcross will then might be call as new mutations. The author say that a single F1 was backcross with MO17 twice (I call MO17-1, the first cross leading to F1, Mo17-2 the first backcross, Mo17-3 the second backcross). We could calculate the probability to have 1 plant only of the 10 plants have the last backcross fragment (MO17-3), it is 0.0097. If we considered the MO17-2, the probability is 0.187, if we considered MO17-1 then it is 0.376. What does it tell us. That if there is some remaining heterozygosity or initial accumulated mutation in the three parental MO17 and the MO17 are slightly different (MO17-1, MO17-2, MO17-3) then SNP with the procedure use by the author might be considered mutation but will be initial heterozygosity or mutations. So there is certainly an overestimation of the mutation rate using the procedure the author used. The author procedure does not allow to completely correcting it. Moreover, the mutation found here also accumulated in the parental inbred if they were “perfect”.

[Response]: We were also aware of the remaining heterozygosity and the possible differences existed in three parental Mo17 could give rise to a bias estimation of mutation rate. To avoid influence of the remaining heterozygosity, we have filtered out the clustered mutations and their flanking variants (please see next response for more details). The slightly difference between Mo17-1, Mo17-2 and Mo17-3 would result in the mutations occurring in two or more individuals, and we also have filtered them out. We must acknowledge that the filtration might not be clean and overestimate the mutation rate. We have added more discussions about the mutation section. Moreover, the genomic fragments with mexicana ancestry do come all from the same genome copy in a single individual, and they are expected to experience the same mutation rate than the Mo17 fragments along the 10 generations of inbreeding. So a large mexicana fragment on chromosome 7 (38,378,754-105,766,898 bp) which were covered by 5 TM individuals was selected to estimate the mutation rate, and the fragment (dash lines) were shown in the following figure:

Using the same filtering criterions, we estimated the mutation rate of mexicana was 2.17×10^{-8} (73 point mutations in a 67,388,144 bp region) which was indeed slightly

lower than 3.87×10^{-8} estimated in the whole Mo17 genome. We have added this result to the revised manuscript. Thank you for your constructive comments.

Q30: L255. If clustered mutation might not be uncommon, it also reflects remaining heterozygosity.

[Response]: We totally agree with your comments. During handling the data, we also noticed the clustered mutation was not uncommon and reflects the remaining heterozygosity. But we have excluded the clustered mutations from further analysis (together with 500 Kb before and after the possible residual heterozygosity region).

The clustered mutations (red dots) could be caused by possible residual heterozygosities. These clusters were excluded from further analysis (together with 500 kb before and after the possible residual heterozygosity region). Only the results of TM3 is shown.

Q31:L219-237. So the mutation rate is estimate through the lence of mapping. Is there difference of mapping quality/depth/ uniqueness between gene and intergenic? Does the higher mutation in gene reflect mapping differential?

[Response]: This is a very good technical question. To check the difference of mapping quality/depth/ uniqueness, we randomly selected 100,000 sites from genic and intergenic regions respectively. We found that the intergenic region tended to have slightly lower mapping depth, mapping quality score and unique mapping rate (genic region: 49%, intergenic region: 42%), that might have little chance to affect mutation identification.

(a) The depth distribution in genic and intergenic region; (b) The mapping quality score proportion in genic and intergenic region. We separate the mapping quality score into 1-49 and 50-60 two categories, because the quality of supporting reads for mutations was ≥ 50 .

Reviewer #4 (Remarks to the Author):

In this article, the authors assemble the draft genomes of the Mo17 variety of maize and the teosinte. On the bases of the two assemblies, they identify and characterize several presence-absence variants. As part of the process, they genotype a set of recombinant inbred lines. With this information, they confirm a polymorphic chromosomal inversion on chromosome 9, they search for QTL for some morphological traits, and they also estimate the spontaneous mutation rate during the 10 generations of inbreeding. This is quite an impressive amount of work, and some interesting results, that could become a beautiful publication if some weak points were addressed before. In what follows, I will make some general recommendations, and more detailed ones on the topic of the mutation rate.

Q32: The main problem I see with the manuscript is a formal one. It is not well structured, what makes it very difficult to understand.

[Response]: Thank you for the reviewer for pointing out this problem. We have re-organized the revised manuscript.

Q33: The introduction should include all the references to the state of the art that are currently cited along the results. Otherwise, it is sometimes difficult to distinguish the original contributions from previously published results. The introduction should also set the goals and motivate the methods. Unfortunately, no justification is given for the most important and original method used: the meta-assembly pipeline. Only in the Discussion we can read that the new genetic design "eliminates assembly problems associated with the heterozygosity of the mexicana parent". However, an evident and simpler approach would have been to inbreed the teosinte and sequence the two genomes separately. Any reader would wonder why the authors mixed up two genomes if they wanted to assemble them separately.

[Response]: Thank you for the reviewer for the constructive comments. We have revised the introduction section and added the goals and motivation of the paper. The

genetic design not only can eliminate assembly problems associated with the heterozygosity of the *mexicana* parent, but also can reduce the number of chimeric contigs caused by the high TE content in maize genome.

Q34: The second problem is that some methods are hardly described. This stems from the excess of results reported. A sharper focus on a few interesting results would improve this aspect, as well as the previous one. For example, the anchoring and ordering of scaffolds with the genetic map of the TM population is not described in the Online methods section. Another example: I did not find an explanation of how the mutations are classified as 'deleterious' or not.

[Response]: We have added the anchoring and ordering of scaffolds in the Online methods. Deleterious mutations were predicted by PROVEAN (**P**rotein **V**ariation **E**ffect **A**nalyzer), and the reference was cited in the corresponding sentence.

Q35: My third main concern is the adequacy of the methods. First, the meta-assembly pipeline does not seem to be the best option to assemble genomes characterized by high structural variation. Even the linkage maps can be tricked by structural variants. The very definition of the genomic bins that are assigned to one parental genome or the other in each TM individual sequenced is based on the coordinates of the B73 genome. The fact that two reads of the same TM individual map to the same bin certainly indicate that they come from the same parental genome, but it does not tell us much about the positions in the parental genome from which they come, unless we assume co-linearity between the genomes of B73, Mo17 and *mexicana*. It is not clear to me to what extent the assemblies depend on this assumption. Plus, the long reads coming from a TM individual help assemble the genome of that individual, which may differ structurally from any of the parentals. In summary, I need an explanation of how the structural variation between Mo17 and teosinte would affect the meta-assembly approach. This is critical, since the validity of the PAVs identified depends on it. To put it another way, despite the efforts to validate the assemblies, I am not convinced that the quality of the assembly is enough to call PAVs.

[Response]: The meta-assembly pipeline contains three major strategies. For the regions with low structure variations, the B73 reference-based assembly was used. For the unmapped reads, we *de novo* assembled them. In addition, we *de novo* assembled 10 individuals, and merged and the above three kinds of contigs, and then connected or extended with PacBio long reads. Finally, we integrated the NRGene scaffolds. As pointed out by the reviewer, two reads of the same TM individual map to the same bin certainly indicate that they come from the same parental genomes, and their positions in the parental genome can be inferred from the combination of bins in different individuals, or the co-linearity between the genomes of B73, Mo17 and *mexicana*. Some of the long reads coming from TM individual may differ structurally from any of the parents, but the major of PacBio reads from TM individual is the same as their parents. In summary, the structural variations (SVs) between Mo17 and teosinte can affect the orientation of scaffolds, but have little effects on the meta-assembly approach, because most of the SVs were assembled with *de novo*

assembly.

To further validate the genome assembly, we sequenced some Mo17 BACs using PacBio technology and compared them with the genome assembly of Mo17. It is observed that these BAC sequences showed >99.5% identity with our Mo17 assembled genome, indicating that the assembled quality is acceptable.

Q36: The introgression analysis is also questionable. It is a very arbitrary heuristic that should be better justified, or changed to match the state of the art.

[Response]: We are sorry for the shortcomings about introgression analysis in previous manuscript. We have revised this part in the following aspects: 1) We described more clearly about the rIBD method and the distribution of the statistic rIBD was shown.

The dash line indicated the threshold ($\frac{(rIBD-\mu)}{\sigma*rIBD} > 2$), the regions on the right were the final candidate introgression regions.

2) As expected the highland maize should have higher introgression rate than others. As Reviewer 3 suggested we re-analyze the correlations between introgression rate and elevation which also provided an indirect evidence for the reliability of rIBD method. As data of the hapmap2 or 3 were collected from numbers of colleges and institutions and the living environment information of the germplasms were not offered in their articles. So we used the other dataset (SNPs number: 946,072) offered by a flowering-time adaptation study in maize landraces (Romero et al., *Nat Genet*, 49, 476-480, 2017; GBS imputed markers at: <http://hdl.handle.net/11529/10035>). 2,900 landraces were detailed with location information (passport data can be accessed upon registration at <http://germinate.seedsofdiscovery.org/maize/>). Using the same introgression detection method, we found that maize from highland Mexico indeed have significantly higher introgression proportion than that in low-elevation (Figure *b, P -value=0.002). Furthermore, we found no matter where the maize came from (including Mexico, Honduras, Guatemala, Panama Costa Rica et al.), the introgression proportion was significantly correlated (P -value=9.703e-06) with the elevation (n=482). This result indicated that some introgression region related to highland adaptation had been kept when the maize expands to other places.

Supplementary Figure 7. a. the positive correlation between elevation and introgression proportion. b. the introgression proportion had significantly difference between highland Mexico maize and lowland maize (low-elevation: <1,200 m above sea level; high-elevation: >1,900 m above sea level)

Q37: My doubts about the adequacy of the methods bring me to the topic of the mutation rate. The ten generations of selfing of the backcrossed population allowed for the accumulation of mutations, that the authors identify in the Mo17 regions as variants represented in a single individual, either heterozygous or homozygous. The authors report a higher mutation rate than previous estimates, despite having filtered out genomic regions suspected to contain ancestral heterozygosity. The authors are aware that the genomic fragments of Mo17 ancestry in the recombinant inbred lines do not come all from the same individual, but from three of them. Given the high estimate, it is reasonable to wonder if the mutation rate is biased by existing variation among the three Mo17 genomes introduced in these lines, despite of the measures taken to prevent this bias. I suggest one way to validate their estimate. The genomic fragments with mexicana ancestry do come all from the same genome copy in a single individual, and they are expected to experience the same mutation rate than the Mo17 fragments along the 10 generations of inbreeding. Despite a lower coverage, that prevents an accurate estimate of the mutation rate from the mexicana fragments, it should be possible to compare the number of mutations observed in the mexicana fragments that are covered in two or more TM individuals with the expectation under the mutation rate estimated in the Mo17 fragments.

[Response]: Thank you for your constructive comments. A large mexicana fragment on chromosome 7 (38,378,754-105,766,898 bp) which were covered by most 5 TM individuals were selected to estimate the mutation rate, and the fragment (dash lines) were shown in the following picture:

Using the same filtering criteria, we estimated the mutation rate of *mexicana* was 2.17×10^{-8} (73 point mutations in a 67,388,144 bp region) which was indeed slightly lower than 3.87×10^{-8} estimated in the whole Mo17 genome. We acknowledged the mutation rate estimated by Mo17 fragments could be higher which could be caused by possible differences between the three Mo17 genomes and have revised the results and descriptions in mutation section.

REVIEWERS' COMMENTS:

Reviewer #1 (Remarks to the Author):

I want to thank the authors for doing a thorough job of responding to comments. From my perspective, the only comment that still gives pause is the issue of positive selection. Again, it bears noting that the different genes for the trio (mex, B73, Mo17) will have variable phylogenetic histories, due to lineage sorting. Indeed, recombination ensures that different parts of genes often do not have the same phylogenetic histories. The problem with this fact is that it can mislead positive selection (PAML-type) approaches, which force a phylogeny. In my opinion, two things could be done in response. One would be simply insert a caveat similar to the arguments I've suggested above. The other is to delete the section. I leave it to the authors.

There are many small grammatical errors throughout. Hopefully Nature Communications will help somewhat.

This manuscript represents a great deal of work, for which the authors should be commended.

Reviewer #3 (Remarks to the Author):

Review of the article of Yang et al.

At this stage the author has answers the main questions I was asking in my review. They add new analysis about introgression and acknowledge partial assembly of teosinte versus maize new genome.

I agree for the publication of this version of the paper.

Sincerely

Reviewer #5 (Remarks to the Author):

The manuscript by Yang et al. has addressed most questions raised by the reviewers.

Some points however remain:

R1Q3: PAVs are explained but in the cases where a region is missing in Mo17 this could be due to a problem in the Mo17 assembly based on uneven coverage. (the method would not call regions not assembled in the Mo17 assembly which is very good) however given the sequence bias of Illumina and a very low PacBio coverage a theoretical possibility is low coverage due to sequence bias. As the regions called are very large so I deem this very unlikely but hinting in this direction somewhere in a few words in the discussion might be helpful

It was a good decision to remove the Mexican genome however this was of course the most interesting genome to be covered here.

R2 Introduction:

I agree that the problems of NRGene including their proprietary and not peer reviewed algorithm is overcome.

Q18:

I am afraid I still see a lack of biology as the JA enrichment is interesting but should be discussed in more detail

R3

Q22 I think the reviewers did what they could and BUSCO yield generally reliable results

R4

Q35: The method the authors propose is indeed interesting and would provide a way around NRGENE (see above) however I still feel the method would need a bit more polishing /explanation especially considering the Mexicana genome

Novel:

I think the manuscript has been greatly improved, however the most interesting piece of work for the community would have been a Mexicana genome of decent quality, which turns out to be almost gene complete but in a generally not so good state.

That one can assemble maize genomes (albeit not as well as the authors) has been shown by Hirsch *Plant Cell*. 2016 Nov; 28(11): 2700–2714. Albeit only reaching an N50 of 0.65Mb. (which is much worse than the 3MB reached for Mo17 but much better than Mexicana.)

Reviewers' comments:

Reviewer #1:

I want to thank the authors for doing a thorough job of responding to comments. From my perspective, the only comment that still gives pause is the issue of positive selection. Again, it bears noting that the different genes for the trio (mex, B73, Mo17) will have variable phylogenetic histories, due to lineage sorting. Indeed, recombination ensures that different parts of genes often do not have the same phylogenetic histories. The problem with this fact is that it can mislead positive selection (PAML-type) approaches, which force a phylogeny. In my opinion, two things could be done in response. One would be simply insert a caveat similar to the arguments I've suggested above. The other is to delete the section. I leave it to the authors.

There are many small grammatical errors throughout. Hopefully Nature Communications will help somewhat.

This manuscript represents a great deal of work, for which the authors should be commended.

[Response]: Thanks so much for your approval to our paper. Taking the editor's and your suggestions, we have deleted the PSG section in the revised manuscript.

Reviewer #3 (Remarks to the Author):

Review of the article of Yang et al.

At this stage the author has answers the main questions I was asking in my review. They add new analysis about introgression and acknowledge partial assembly of teosinte versus maize new genome.

I agree for the publication of this version of the paper.

Sincerely

[Response]: We are grateful for your approval to our paper and your professional suggestions.

Reviewer #5 (Remarks to the Author):

The manuscript by Yang et al. has addressed most questions raised by the reviewers.

Some points however remain:

R1-Q3: PAVs are explained but in the cases where a region is missing in Mo17 this could be due to a problem in the Mo17 assembly based on uneven coverage. (the method would not call regions not assembled in the Mo17 assembly which is very good) however given the sequence bias of Illumina and a very low PacBio coverage a theoretical possibility is low coverage due to sequence bias. As the regions called are

very large so I deem this very unlikely but hinting in this direction somewhere in a few words in the discussion might be helpful.

It was a good decision to remove the *mexicana* genome however this was of course the most interesting genome to be covered here.

[Response]: Indeed, the sequence bias of Illumina and the low coverage of Pacbio sequences could overestimate the number of PAVs. As your suggestion, we have added some hints in discussion (Line 282 in the related manuscript).

R2-Introduction:

I agree that the problems of NRGene including their proprietary and not peer reviewed algorithm is overcome.

[Response]: Thanks for your comments. The proprietary and algorithm of NRGene should not be problems, and two papers using NRGene technology have been published: Avni R et al., Science, 2017 and Hirsch et al., Plant Cell. 2016.

R2-Q18:

I am afraid I still see a lack of biology as the JA enrichment is interesting but should be discussed in more detail.

[Response]: Thanks for the reviewer for the constructive comments. However, according to the suggestions from editor and reviewer 1, we have deleted the PSG section in the revised manuscript. We have identified many QTLs by using the BC2F7 populations and some of them were closed to be cloned and hope we can present some interesting biology in the near future. In the present manuscript, we have confirmed the ~27 Mb inversion *Inv9d* was real by the linkage map and the *Inv9d* showed altitudinal clines in environmental association analysis. In our studies, a large QTL for ear leaf width was identified in *Inv9d* region indicating the *Inv9d* not only contributed to adaptation but also plant morphology. Through RNA-seq analysis of the 10 selected TM individuals, we observed no different expression pattern in *Inv9d* region. It's also hard to say the *Inv9d* or a gene in *Inv9d* controlled the adaptation and plant morphology, because the rare recombination in this inversion hindered the QTL fine-mapping. However, it has provided a good example that the inversion region may associate with the maize adaptation.

R3-Q22 I think the reviewers did what they could and BUSCO yield generally reliable results

[Response]: Thank you for the praise.

R4-Q35: The method the authors propose is indeed interesting and would provide a way around NRGENE (see above) however I still feel the method would need a bit more polishing /explanation especially considering the Mexicana genome

[Response]: Thank you for the approval to our method. To explain the meta-assembly pipeline more clearly, we added more details especially for the method of merging

contigs used for assembling the *mexicana* and Mo17 contigs. The revised content has been labeled in red in the supplementary note. We hope these efforts could meet your requirement.

Novel:

I think the manuscript has been greatly improved, however the most interesting piece of work for the community would have been a Mexicana genome of decent quality, which turns out to be almost gene complete but in a generally not so good state.

That one can assemble maize genomes (albeit not as well as the authors) has been shown by Hirsch *Plant Cell*. 2016 Nov; 28(11): 2700–2714. Albeit only reaching an N50 of 0.65Mb. (which is much worse than the 3MB reached for Mo17 but much better than Mexicana.)

[Response]: Thank you for the praise. We agree that there is still large space to improve the quality in the future.